# Dual-resolving of positional and geometric isomers of C=C bonds via bifunctional photocycloaddition-photoisomerization reaction system

Guifang Feng[1], Ming Gao[1], Liwei Wang[1], Jiayi Chen[2], Menglu Hou[1], Qiongqiong Wan[1], Yun Lin[2], Guoyong Xu [1], Xiaotian Qi [3] & Suming Chen [1✉]

The biological functions of lipids largely depend on their chemical structures. The position and configuration of C=C bonds are two of the essential attributes that determine the structures of unsaturated lipids. However, simultaneous identification of both attributes remains challenging. Here, we develop a bifunctional visible-light-activated photocycloaddition-photoisomerization reaction system, which enables the dual-resolving of the positional and geometric isomerism of C=C bonds in lipids when combines with liquid chromatography-mass spectrometry. The dual-pathway reaction mechanism is demonstrated by experiments and density functional theory calculations. Based on this bifunctional reaction system, a workflow of deep structural lipidomics is established, and allows the revealing of unique patterns of *cis-trans*-isomers in bacteria, as well as the tracking of C=C positional isomers changes in mouse brain ischemia. This study not only offers a powerful tool for deep lipid structural biology, but also provides a paradigm for developing the multifunctional visible-light-induced reaction.

[1] The Institute for Advanced Studies, Wuhan University, Wuhan, Hubei 430072, China. [2] Department of Anesthesiology, Union Hospital, Tongji Medical College, Huazhong University of Science and Technology, Wuhan, Hubei 430022, China. [3] Engineering Research Center of Organosilicon Compounds & Materials, Ministry of Education, College of Chemistry and Molecular Sciences, Wuhan University, Wuhan, Hubei 430072, China. ✉email: sm.chen@whu.edu.cn

ipids are a group of biomolecules that play crucial roles in organisms, such as the structural elements of biological membranes, energy storage, and signal transduction[1,2]. Unsaturated lipids are a subclass of lipids containing either a single or multiple C=C bonds in the fatty acyl chains[3,4]. The types of isomers of C=C bonds in lipids includes positional and geometric (*cis-trans* or *E-Z*) isomers, which endow lipids with different structures and functions in living organisms[5–7]. For instance, C=C positional isomer ratio shows significant correlations to the onset/progression of breast cancer[8], and lipid acyl chain *cis*-C=C bond position was found to modulate plasma membrane domain registration/anti-registration[9]. The geometry of the double bonds in naturally occurring fatty acid (FA)-containing lipids of eukaryotes is predominantly *cis* type. Notably, researches have implicated *trans* FAs as being particularly deleterious to human health[10], contributing to problems such as cardiovascular diseases[11], systemic inflammation[12], and more recently hepatic diseases[13]. However, *trans* FAs are found closely related to the activity and biofunction of bacteria. For example, *trans* unsaturated FAs were known as the by-products of FA transformations carried out by the obligate anaerobic ruminal microflora. They are also the critical membrane constituents of various bacteria, mainly Gram-negative bacteria. The *trans* isomers in bacteria could be synthesized by a direct isomerization of the complementary cis-configuration of the double bond[14]. Interestingly, the positions of the C=C of microbial C18-FAs depend on the biosynthetic route by which they are produced; whereas during the anaerobic pathway leads to *cis*-vaccenic acid [C18:1 (Δ11)], the aerobic pathway produces oleic acid [C18:1 (Δ9)][14]. The *cis-trans* and positional conversion of C=C bonds in unsaturated lipids changes the phase-transition temperatures, rigidity, and permeability of the membrane of bacteria, which are critical to their survival in response to environmental stimuli[2]. Thus, the simultaneous resolving of C=C location and *E-Z* isomers of unsaturated lipids could provide multidimensional information for in-depth understanding of the relationship between their structures and biological roles[6].

The reliable resolving of both C=C positional and geometric isomers remains challenging, even though efforts have been made on the analyses of single isomeric type. Mass spectrometry (MS) has played important roles in the analysis of the C=C bonds isomers due to its high sensitivity and structural elucidation capabilities[5,15–17]. For the identification of the location of C=C bonds, some special methods for inducing dissociation at the C=C bonds in MS were developed[18–21]. Meanwhile, selective chemical derivatizations or activation of C=C bonds prior to MS analysis also emerged recently[5,15,17,22–25]. Despite the progresses, these methods lack the capability for simultaneous identification of the position and geometric configurations of C=C bonds. In a way, the reliable identification of configuration of C=C bond is even more difficult[6,26,27]. Although the liquid chromatographic (LC) and ion mobility spectrometry (IMS) methods may separate the *E/Z* isomers and provide information by matching their elution times or arrival times with that of the known standard lipids[27–29], the acquirement of standards for numerous lipids in complicated biological systems is quite difficult. The greater difficulty is to confirm the lipid C=C configuration of a single peak in LC when only one kind of the *cis*/*trans* isomer is present. In this case, no reference information of a counterpart isomer is available. In addition, the varied LC and IMS conditions may also confuse the assignment of these isomers. Besides the MS methods, high resolution nuclear magnetic resonance (NMR) spectroscopy was also used to identify the C=C bond isomers[30]. For example, a band-selective two-dimensional (2D) $^1$H-$^{13}$C heteronuclear single quantum coherence (HSQC) experiment was utilized to identify the terminal methyl protons of minor *trans*

FAs[31]. The 2D HSQC-total correlation spectroscopy experiment could correlate carbon signals with the adjacent allylic protons of unsaturated FAs and analyze the positional distribution of unsaturated FA chains[32]. Furthermore, a semi-selective $^1$H–$^{13}$C HSQC experiment was developed to obtain ultra-high resolution data in the $^{13}$C double bond region, and to assign and quantify individual unsaturated FAs in biological samples[33]. Nevertheless, the comprehensive identification of micro-amount lipid C=C isomers other than FAs in complicated samples with NMR techniques remains a challenge. Due to the essential need for elucidating the structures of lipids C=C isomers[6], we are aiming to develop a bifunctional strategy that could simultaneously analyze locations and *E-Z*-configurations of C=C bond, and provide multidimensional isomeric information for its biological function study.

Triplet-energy transfer from the photocatalyst to the carbonyl substrates was reported that could trigger the [2 + 2] cycloaddition reaction of carbonyls and alkenes via visible-light[34,35]. This approach negates the need for both visible-light-absorbing carbonyl substrates and UV light to enable access to a variety of functionalized oxetanes. We reason that this unique energy transfer-based chemistry should also be extended to selective C=C bonds derivatization in lipids, of which the locations could be identified based on the structural analysis of the fragments generated from formed oxetane products when coupled with tandem MS[5]. More importantly, we envision that this process may also induce the photoisomerization of C=C bonds, which was not reported for this reaction system. In fact, the sensitized photoisomerization activated by visible light via direct energy transfer or addition−elimination of ketone to the C=C bond[36] was proved feasible[37,38]. Therefore, the selective photoisomerization tendency from the less thermodynamically stable *cis* lipid isomers to more thermodynamically stable *trans* isomers could provide a reliable dimension to discriminate them when combined with LC-MS. The configurations of C=C bonds could be accurately identified by comparing their LC patterns before and after the photoisomerization reaction. Thus, the same photocatalytic reaction system should enable the bifunctional identification of both the location and *cis-trans* configurations of C=C bonds in unsaturated lipids.

Herein, we established a visible-light-activated photocatalytic reaction system using methyl benzoylformate (MBF) as carbonyl substrate and Ir[dFppy]$_2$(dtbbpy)PF$_6$ as photocatalyst for unsaturated lipids (Fig. 1). This bifunctional system features: (1) high-efficient [2 + 2] photocycloaddition derivatization of C=C bonds in unsaturated lipids; (2) photoisomerization conversion of *cis*- to *trans*-C=C bonds. The possible mechanism of the photocycloaddition-photoisomerization (PCPI) reaction was studied by experiments and density functional theory calculations. By using this reaction system, we established an integrated workflow that enables the comprehensively qualitative and quantitative analysis of the C=C bonds isomers of lipids with LC-MS. The applicability of this method was validated by deeply analyzing lipid structures of bacteria and revealing their unique patterns of *cis-trans*-isomers, as well as tracking the C=C positional isomers changes in brain ischemia with a mouse model.

## Results and discussion

**Design and validation of the bifunctional photocatalytic reaction system.** Direct visible-light activated [2 + 2] cycloaddition reaction of anthraquinone or benzophenone and unsaturated lipids coupled with tandem MS analysis has exhibited capability for locating the C=C bonds[5,25]. However, the photoisomerization of C=C bonds was not reported, and the reaction yields are

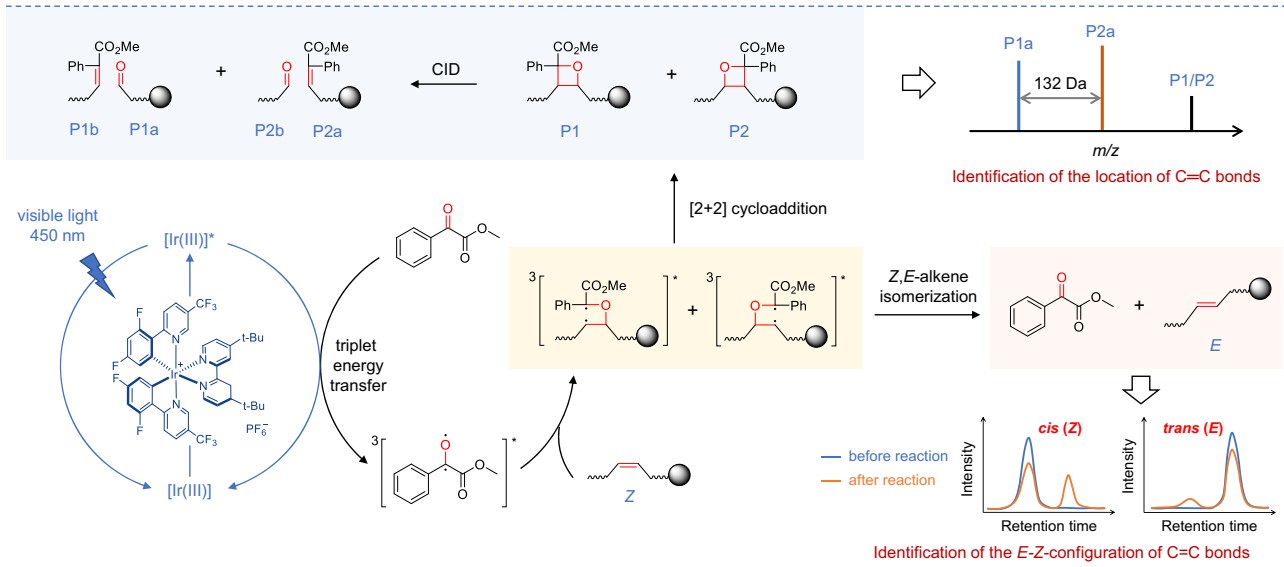

**Fig. 1 Schematic illustration of the bifunctional visible-light-activated photocatalytic reaction system.** For the analysis of C=C bonds positional isomer, methyl benzoylformate was excited to triplet by the photoexcited Ir[dF(CF₃)ppy]₂(dtbbpy)PF₆ via triplet-energy transfer. Then it underwent subsequent [2 + 2] cycloaddition with C=C bonds in lipids via the standard stepwise biradical mechanism of the Paternò−Büchi reaction to afford oxetanes P1 and P2. The oxetane isomers could be fragmented to the diagnostic ions P1a, P1b, P2a, and P2b to indicate the locations of C=C bonds via collision-induced dissociation (CID) in tandem MS. For the analysis of E-Z isomer, the (Z)-isomer of lipids prefers to be converted to (E)-isomer via the triplet-energy transfer from the excited photocatalyst Ir[dF(CF₃)ppy]₂(dtbbpy)PF₆ in the same reaction system.

compromised by the relatively low absorption efficiency of visible light. We reasoned that the higher efficient excitation of carbonyl substrate to triplet state would increase the yield. Inspired by the photocatalytic triplet-energy transfer[39], we consider the transition metal photocatalysts that could efficiently absorb visible light and be excited to a long-lived triplet state can serve as an efficient energy transfer mediator to productively activate carbonyl substrates[39]. After evaluating the efficiencies of different reaction systems with unsaturated lipids, Ir[dF(CF₃)ppy]₂(dtbbpy)PF₆ (triplet energy ~60.1 kcal/mol) and MBF (triplet energy ~60.0 kcal/mol) with close proximity triplet energy were chosen as photocatalyst and carbonyl substrate to facilitate the efficient energy transfer (Fig. 1). The reasonable mechanism may start from the excitation of Ir[dF(CF₃)ppy]₂(dtbbpy)PF₆ to triplet state by visible light. This fluorinated Ir(III) complex with high triplet energy would promote the generation of long-lived triplet [MBF]* via Dexter energy transfer[34,39]. Then, ³[MBF]* would undergo subsequent [2 + 2] cycloaddition with C=C bonds in lipids via the standard stepwise biradical mechanism of the Paternò−Büchi reaction to afford oxetane[36,40]. Especially, the ³[MBF-lipid]* biradical intermediate may also undergo radical elimination to regenerate ground state MBF and geometrically isomerized lipids[36].

To confirm the above mechanism, the density functional theory (DFT) calculations were conducted to investigate the visible-light-activated photocatalytic reaction of oleic acid [FA 18:1 (9Z)] and MBF (Fig. 2, the cartesian coordinates and energies of optimized structures were shown in Supplementary Information). Computational results suggested that the initial radical addition of biradical intermediate **2** towards oleic acid (**1**) occurs easily through **TS-1** (ΔG‡ = 3.7 kcal/mol) and **TS-2** (ΔG‡ = 3.8 kcal/mol), leading to the formation of a biradical isomeric intermediate **3** and **3a**, respectively. The C–C rotation in **3** and **3a** can generate the conformers **4** and **4a**. Subsequent β-scission of carbon-centered radicals **4/4a** (via **TS-3** and **TS-4**) would accomplish the Z-to-E-alkene isomerization with an activation free energy of 15.2 and 16.3 kcal/mol. It is shown that

transition states **TS-1/TS-2** and **TS-3/TS-4** have comparable energies. Moreover, the generated elaidic acid **5** is 1.1 kcal/mol more stable than oleic acid **1**, which indicates the Z,E-alkene isomerization is kinetically and thermodynamically feasible. Besides, the intramolecular radical-radical cross-coupling in **3** and **3a** can favorably form the four-membered [2 + 2] cycloaddition products **6** and **6a** (ΔG = − 59.6 and −59.1 kcal/mol).

The experimental results also confirmed the photocycloaddition and photoisomerization reaction pathways. As a starting point for this investigation, a pair of monounsaturated fatty-acid isomers FA 18:1 (9Z/9E) were subjected to this photocatalytic reaction (Fig. 3a). As shown in Fig. 3b, the reaction gave the [2 + 2] cycloaddition products P1/P2 in a high relative ion intensity (RI) of 72% (RI = $I_{product}/(I_{product} + I_{reactant})$) using 5 mol% photocatalyst Ir[dF(CF₃)ppy]₂(dtbbpy)PF₆ and 5 equivalent of methyl benzoylformate after 10 min reaction in a glass vial (1.5 mL) irradiated by a blue LED (40 W). The two isomeric oxetanes of P1 and P2 ([M + Na]⁺ at m/z 469.29) could be fragmented by collision induced dissociation (CID) in tandem MS analysis, and the diagnostic ions P1a (consisting of an aldehyde and head group) and P2a (consisting of methyl 2-phenylacrylate and head group) at m/z 195.10 and 327.16 were detected, respectively (Fig. 3c). These characteristic fragments provided not only unambiguous evidence of the formation of [2 + 2] photocycloaddition products but also diagnostic ions for the identification of the position of C=C bonds. The ion pair P1a and P2a has a mass difference of 132 Da (C₉H₈O) and could be simply used as "diagnostic ions" to confirm the presence of C=C bonds. The neutral loss of fragment P1b (consisting of methyl 2-phenylacrylate) and P2b (consisting of an aldehyde) (Fig. 1) was not detected (Fig. 3c), but their formulas and degrees of unsaturation could be a direct indication of C=C bond positions. This visible-light-activated photocatalytic reaction represents a type of chemical method for identifying the location of C=C bond. More importantly, we found the high tendency of the conversion from cis-C=C bond to its trans- counterpart in the same photocatalytic reaction system (Fig. 3d). In the extracted ion

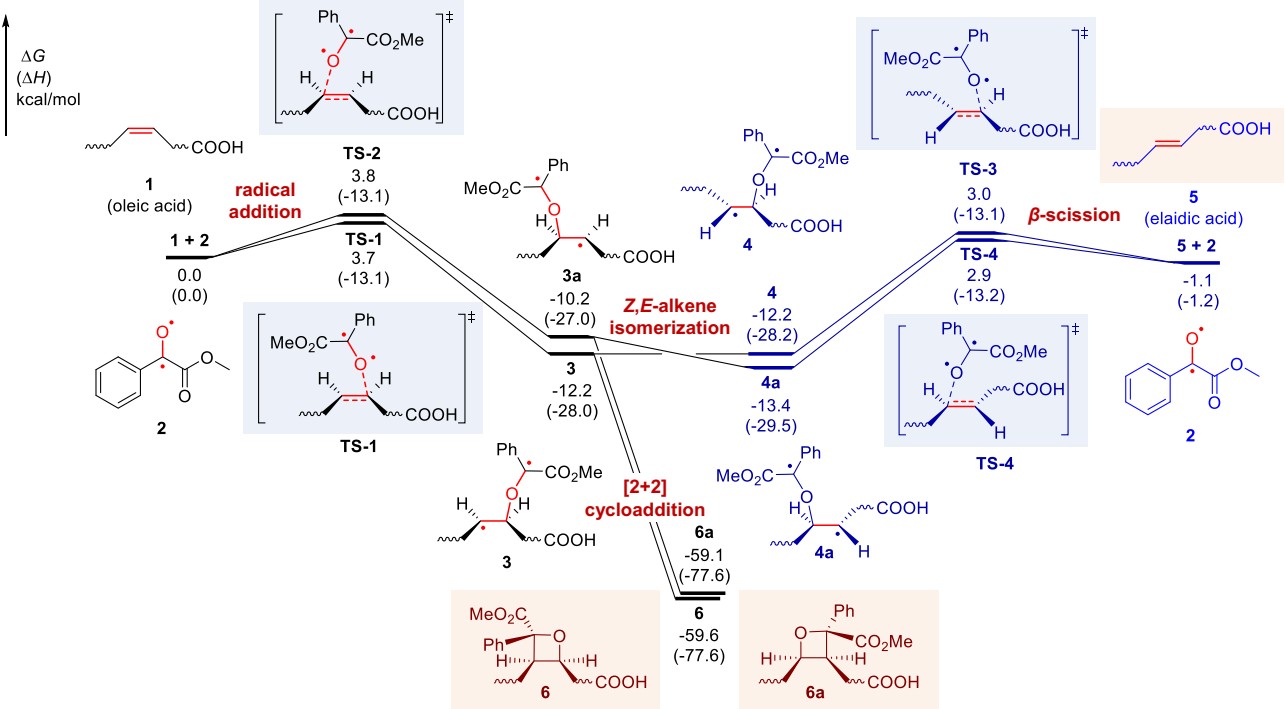

**Fig. 2 Free-energy profile of the visible-light-activated photocatalytic reaction of oleic acid and methyl benzoylformate (MBF).** Density functional theory calculations were performed at M06-2X/6-311 + G(d,p)/SMD(acetonitrile)//M06-2X/6-31 G(d) level of theory. The Z,E-alkene isomerization pathway is shown in blue. ΔG, change in Gibbs free energy; ΔH, change in enthalpy.

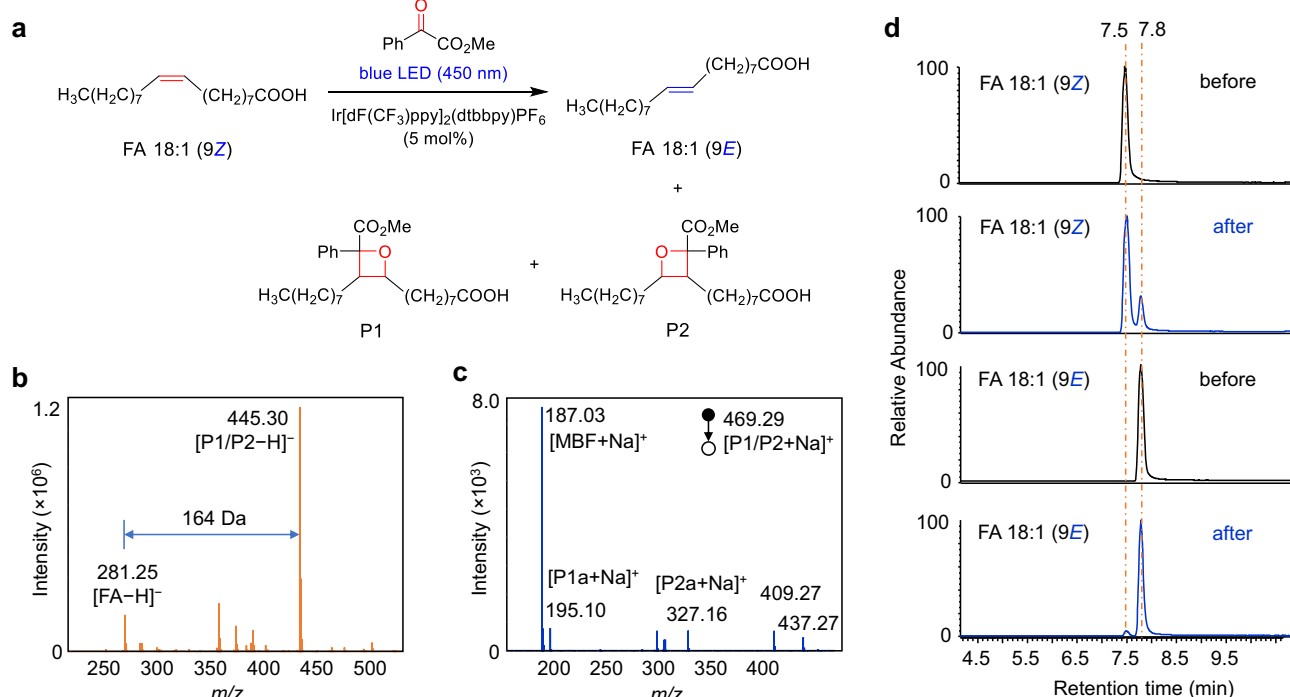

**Fig. 3 Demonstration of the bifunctional reaction system for the identification of C=C bonds location and E-Z configuration in FA 18:1 (Δ9).**
**a** Photocycloaddition and photoisomerization reactions of FA 18:1 (9Z). The photocatalyst is Ir[dF(CF₃)ppy]₂(dtbbpy)PF₆. **b** Mass spectrum of the photocatalytic reaction solution of FA 18:1 (9Z) in negative ion mode. **c** MS/MS spectrum of the sodiated photocycloaddition product P1/P2 in positive ion mode. Peak at m/z 187.03 denotes the fragment ion of the methyl benzoylformate motif. Peaks at m/z 409.27 and 437.27 may be assigned to the fragment ions of P1/P2 after losing the CO₂Me and MeOH structures, respectively. **d** Extracted ion chromatograms (EIC) of FA 18:1 (9Z) and FA 18:1 (9E) before and after the photocatalytic reaction in negative ion mode.

chromatograms (EICs), FA 18:1 (9Z) standard shows a single chromatographic peak at 7.5 min before the reaction. However, an evident new peak corresponds to FA 18:1 (9E) was observed at 7.8 min after the reaction. In contrast, the conversion from FA 18:1 (9E) to FA 18:1 (9Z) was very little (~1.7%). These results revealed the preferred *cis*-to-*trans* photoisomerization of unsaturated FAs in this photocatalytic reaction system, which provided an additional characteristic for identifying the geometric configuration of the C=C bond.

To further distinguish the photoisomerization mechanism from two plausible reaction pathways, i.e., direct energy transfer and addition–elimination, different photoreactions of FA 18:1 (9Z/9E) were conducted (Supplementary Fig. 1). The results showed that no isomerized product of FA 18:1 (9Z) were observed when only Ir[dF(CF₃)ppy]₂(dtbbpy)PF₆ was added (Supplementary Fig. 1a). A slight conversion (ratio of peak area, $R_A$ = 15.3%) of FA 18:1 (9Z) to its 9E-isomer was detected in the FA-MBF reaction system, which might be induced by a small portion of excited MBF due to the existence of partial emission in UV region from LED light (Supplementary Fig. 2). Expectedly, the three-component system of FA-MBF-Ir (III) catalyst gave a much higher ratio of FA 18:1 (9E) ($R_A$ = 29.3%). Likewise, the same trend was observed for the conversion from FA 18:1 (9E) to FA 18:1 (9Z), but the yields were much lower in both FA-MBF ($R_A$ = 1.5%) and FA-MBF-Ir (III) catalyst ($R_A$ = 2.5%) reaction systems (Supplementary Fig. 1b). The failed photoisomerization in FA-Ir (III) catalyst system led us to propose that the direct energy transfer pathway may not account for the isomerization of FA 18:1, since the triplet energy of MBF and Ir (III) catalyst is in close proximity (~60 kcal/mol). Instead, addition–elimination of ketone to C=C should be the predominant mechanism (Fig. 2)[36]. The generation of triplet state MBF could be partially induced by the direct irradiation of LED light, but will be greatly enhanced by the energy transfer from excited Ir[dF(CF₃)ppy]₂(dtbbpy)PF₆. These results provided additional evidences for the mechanism of the proposed bifunctional photocatalytic reaction system.

The unique *cis*-to-*trans* photoisomerization could be utilized to identify the *cis*- or *trans*-C=C bonds in lipids when combined with LC-MS analysis. The C=C bond in a monounsaturated lipid would be identified as *cis*-configuration if the second peak corresponds to its *trans*-isomer appeared with a longer retention time (RT) after the photocatalytic reaction. Nevertheless, the C=C bond would be identified as *trans*-configuration if no isomeric peak with longer RT appeared, and the *cis*-isomer with shorter RT should be very little or even absent. *Cis*-isomers appeared always at a lower retention time than the *trans*-isomers, because the *cis*-fatty acyl moieties experience a weaker interaction with saturated C18 chains of the reversed stationary phase in HPLC than that of *trans*-isomers due to the U-shaped geometry of the *cis*-isomers[41]. For the conventional LC-MS analysis, there is no good way to confirm the configuration of C=C bond in lipids if only separate *cis*- or *trans*- isomer is present. The observation of this photoisomerization would provide reliable confirmation of *cis*- or *trans*- isomers by comparing their LC patterns before and after the reaction.

The similar results were also obtained in the reaction of monounsaturated phosphatidylcholine (PC) 16:0/18:1 (9Z) and 18:1 (9Z)/16:0 (Supplementary Fig. 3). The mass spectrum of the reaction solution showed clear peak of [2 + 2] photocycloaddition product at *m/z* 924.6351, which is just 164 Da more than the original PC 16:0/18:1 (9Z) at *m/z* 760.5884 (Supplementary Fig. 3a). The two diagnostic ions of P1a and P2a observed at *m/z* 782.4976 and 650.4405 respectively from the MS/MS spectrum of product ions (P1/P2) indicated the accurate location of C=C bond (Supplementary Fig. 3b and S3c). In addition, a new peak with longer RT appeared in the EIC of PC 16:0/18:1 (9Z) and 18:1

(9Z)/16:0 after the photocatalytic reaction, which verified the Z-configuration of the C=C bond. Overall, these results showed the preliminary feasibility of the bifunctional photocatalytic reaction system for the concurrent identification of location and *cis*-*trans* configuration of lipids.

**Validation of the bifunctional reaction system with poly-unsaturated lipids**. To further demonstrate the application of this strategy, unsaturated lipids with two C=C bonds were selected. Firstly, the PCPI reaction of FA 18:2 (9Z, 12Z) was investigated. As shown in Fig. 4a, the FA could be efficiently converted to the [2 + 2] cycloaddition products P1/P2. The MS/MS spectrum (Fig. 4b) of [P1/P2 + Na]⁺ generates diagnostic ions of C=C bonds at *m/z* 195.10/327.16 (Δ9) and *m/z* 235.13/367.19 (Δ12). The characteristic mass difference of 132 Da (C₉H₈O) between fragments P1a and P2a further confirmed the structures of photo-cycloaddition product. In addition, the EICs of FA 18:2 (9Z, 12Z) and FA 18:2 (9E, 12E) before and after the photocatalytic reactions showed the different isomerization tendency (Fig. 4e). Two new peaks were observed on the right side for FA 18:2 (9Z, 12Z) after reaction, although the rightmost peak corresponds to the conversion of both two C=C bonds [FA 18:2 (9E, 12E)] was minor. We speculate that the peak in the middle should belong to the mixture of FA 18:2 (9Z, 12E) and FA 18:2 (9E, 12Z), which might hard to be separated due to their much similar structures. Expectedly, the photoisomerization of FA 18:2 (9Z, 11E) also observed with a new peak [FA 18:2 (9E, 11E)] on the right, but the conversion to FA 18:2 (9Z, 11Z) was much less. Thus, the original geometric configurations of FA with two C=C bonds could be identified by comparing the EICs after reaction to that before reaction. In this case, the bis-allylic CH₂ protons are very reactive for hydrogen atom abstraction. One may surmise that the hydrogen atom abstraction pathway might compete with the radical addition to the alkene, which ultimately results in a complicated outcome for the identification of C=C bonds location and *E-Z* configuration. DFT calculations were then performed to study these two competitive pathways using FA 18:2 (9Z, 12Z) as the reactant (Supplementary Fig. 4). The computational results support that the radical addition to alkene via **TS-5** is kinetically favored over the hydrogen atom abstraction via **TS-6**. Therefore, the hydrogen atom abstraction pathway cannot outcompete the radical addition step.

Moreover, polyunsaturated phospholipids such as PC 18:1 (9Z)/18:1 (9Z) and PC 18:0/18:2 (9Z, 12Z) were also investigated. The products of cycloaddition were detected at *m/z* 950.65 for PC 18:1 (9Z)/18:1 (9Z) (Fig. 4c), and the fragment ions of P1a and P2a also appeared at *m/z* 808.51 and 676.45, respectively, in tandem MS (Fig. 4d). Not surprisingly, the products of photoisomerization were clearly observed in EIC, which should correspond to the isomerization of one (the middle peak) and two C=C bonds (the right peak) (Fig. 4f). The rightmost peak could match well with PC 18:1 (9E)/18:1 (9E). Likewise, the locations of C=C bonds in PC 18:0/18:2 (9Z, 12Z) could also be identified by using this visible-light-activated photocatalytic reaction and tandem MS (Supplementary Fig. 5a). Characteristic diagnostic ions P1a and P2a derived from the CID of oxetane units at Δ9 and Δ12 were clearly observed. Two new isomers also appeared in EIC after reaction as to PC 18:0/18:2 (9Z, 12Z) (Supplementary Fig. 4b). These data revealed the unique triple-peak EIC pattern of unsaturated lipids containing two *cis*-C=C bonds after photoisomerization. Furthermore, FA 20:4 (5Z, 8Z, 11Z, 14Z) was used to test the ability of this method to the identification of lipids with complex C=C composition. The MS/MS spectrum of the photocycloaddition products shows clearly four pairs of diagnostic ions, which correspond to the locations of double bonds at Δ5, 8, 11, and 14 (Supplementary Fig. 6a). On the other hand, the EIC of FA 20:4

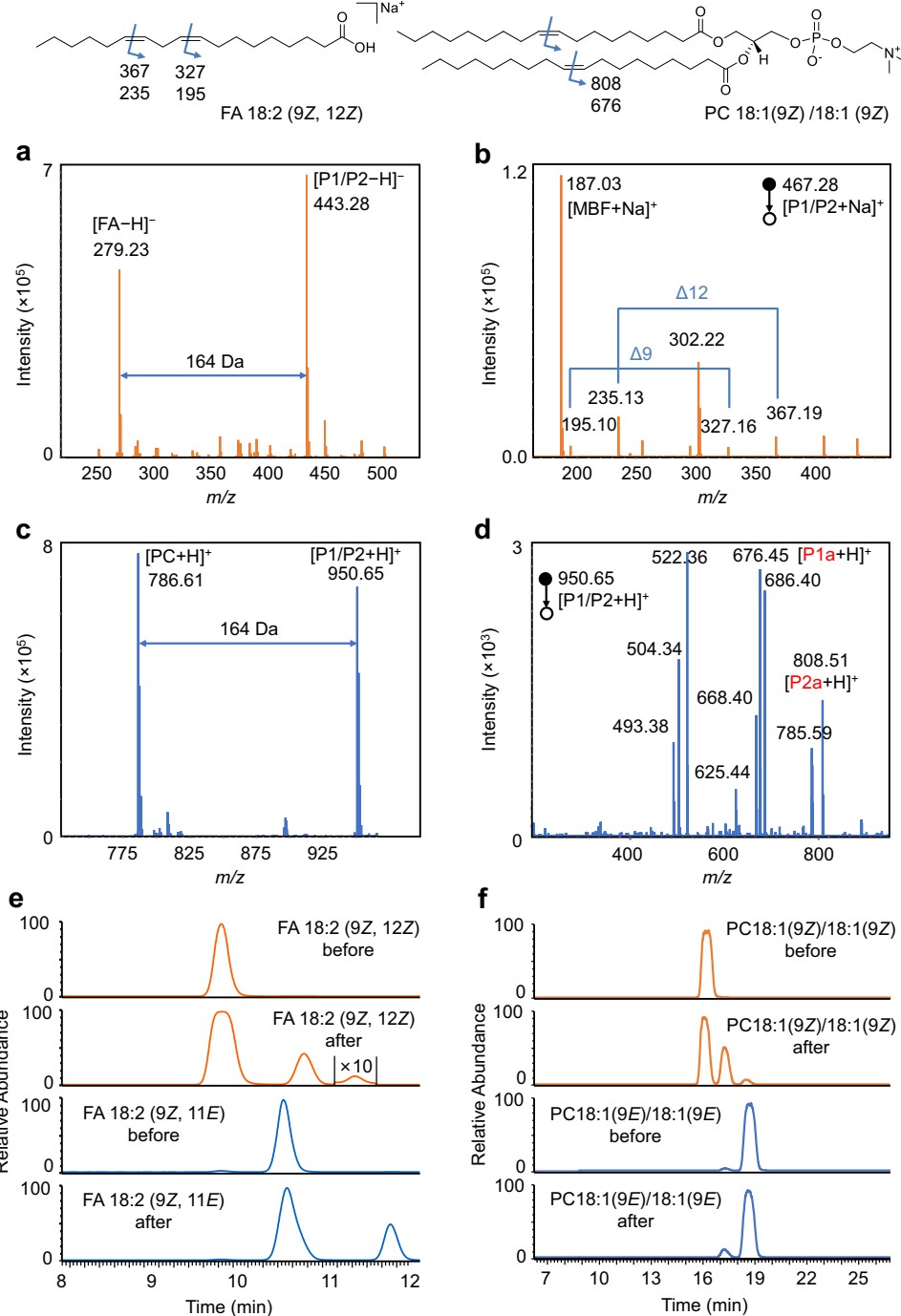

**Fig. 4 Demonstration of the bifunctional reaction system for the identification of C=C bonds location and *E-Z* configuration in polyunsaturated lipids.** **a** Mass spectrum of the photocatalytic reaction solution of FA 18:2 (9Z, 12Z). **b** MS/MS spectrum of the sodiated photocycloaddition product (P1/P2) of FA 18:2 (9Z, 12Z). **c** Mass spectrum of the photocatalytic reaction solution of PC 18:1 (9Z)/18:1 (9Z). **d** MS/MS spectrum of the protonated photocycloaddition product (P1/P2) of PC 18:1 (9Z)/18:1 (9Z). Assignment of the fragments: m/z 808.51, [P1a + H]⁺; m/z 785.59, [PC]⁺; m/z 686.40, [P1/P2 − FA 18:1]⁺; m/z 676.45, [P2a + H]⁺; m/z 668.40, [P1/P2 − FA 18:1 − H₂O]⁺; m/z 625.44, [P1a − C₅H₁₄NO₄P (183)]⁺; m/z 522.36, [PC − FA 18:1]⁺; m/z 504.34, [PC − FA 18:1 − H₂O]⁺; m/z 451.34, [P2a − C₅H₁₄NO₄P (183)]⁺. **e** EICs of FA 18:2 (9Z, 12Z) and FA 18:2 (9Z, 11E) before and after the photocatalytic reaction at m/z 279.2 in negative ion mode. **f** EICs of PC 18:1 (9Z)/18:1 (9Z) and PC 18:0/18:2 (9Z, 12Z) before and after the photocatalytic reaction at m/z 786.6 in positive ion mode.

after photoisomerization showed only the *cis*-to-*trans* conversion, which confirmed its original *cis*-configuration before the reaction (Supplementary Fig. 6b). The above results indicated the abilities of this bifunctional PCPI reaction system coupled with MS in the identification of the location and geometric isomerism for unsaturated lipids. The photoisomerization provides the reliable

identity of C=C bond, which is very difficult to be realized by conventional LC-MS methods.

**Identification of the locations of C=C bonds in glycerides.** Besides the FA and phospholipids, the visible-light-activated photocatalytic cycloaddition reaction of MBF with glycerides was

also carried out and detected by MS (Supplementary Fig. 7). For diacylglyceride (DG) 16:0/18:1 (9Z), the characteristic ions indicated the location of C=C were detected in the form of [P1a/P2a − H2O + H]+, which corresponds to the loss of water from the glycerol head group (Supplementary Fig. 7a). The loss of water was also observed in the MS/MS spectrum of original DG 16:0/18:1 (9Z) (Supplementary Fig. 7b). We then investigated the reactions of triacylglycerides (TG) with MBF. Expectedly, both TG 18:1 (9Z)/18:1 (9Z)/18:1 (9Z) and TG 18:2 (9Z,12Z)/18:2 (9Z,12Z)/18:2 (9Z,12Z) gave good yields of [2 + 2] photocycloaddition products with MBF, and the characteristic ions derived from P1a and P2a by losing one fatty acyl chain could be found in the MS/MS spectrum to pinpoint the C=C bonds (Supplementary Fig. 7c and S7d). These results implied the broad scope of the photocatalytic [2 + 2] cycloaddition reaction for the identification of C=C bonds in lipids.

**Quantitative analysis of the C=C isomers.** The change of the ratio of C=C positional isomers has been found to correlate with the state of diseases[42]. To examine the quantitative performance of the photocatalytic cycloaddition reaction-based MS method for C=C location isomers, the lipids mixtures consist of Δ6 and Δ9 isomers of FA 18:1 (6Z/9Z) and PC 18:1 (6Z/9Z)/18:1 (6Z/9Z) were analyzed. The chromatographic peak areas (A) of diagnostic ions P1a and P2a derived from the cycloaddition products were used for quantitation (Supplementary Fig. 8). The results (Supplementary Fig. 9) showed that both the peak area ratios ($A_{\Delta 9/\Delta 6}$) of individual ions and the sum of P1a/P2a are directly proportional to the molar ratios ($M_{\Delta 9/\Delta 6}$) of the two isomers with good linearities ($R^2 > 0.99$) in a wide dynamic range (1:2 to 20:1), which demonstrated the capability of this approach in quantitative analysis the ratio of C=C location isomers, and would provide the opportunity to reveal the relationship between the lipids C=C isomer compositions and the diseases. On the other hand, the quantification of the C=C geometric isomers could be simply based on the original LC peak areas of each isomer after identifying their configurations.

**Comprehensive identification of lipid structures in bacterial samples.** Bacterial survival depends on the homeostasis of membrane lipid and on the ability to adjust lipid compositions to acclimatize the bacterial cell to different environments. The lipid homeostasis in bacteria, such as cis-trans and C=C isomerization, is crucial to maintain their biofunctions[2,14,43]. Therefore, comprehensive identification of lipids at an isomeric level is required. To this end, we established a workflow of deep structural lipidomics by utilizing the bifunctional visible-light-activated PCPI reaction system and LC-MS/MS (Fig. 5a). In this approach, the information of lipid subclasses and fatty acyls was first identified by LC-MS/MS and MS-DIAL software by matching the MS and MS² spectra with local library of LipidBlast. Then, the extracted lipids were reacted with MBF under the photocatalysis of Ir(III) complex catalyst. The C=C geometric configurations of unsaturated lipids were confirmed by comparing the EIC patterns of photo-isomerized products with that of original lipids. On the other hand, the locations of C=C bonds was obtained by analyzing the MS/MS spectrum of the photocycloaddition products. This approach was applied to the deep lipidomic analysis of four bacterial strains. Among them, P. syringae is a pathogenic bacterium of Arabidopsis, while the other three (i.e., W. ginsengihum, P. citronellolis, and M. osloensis.) are endophytic bacteria isolated from Arabidopsis leaves. Take the analysis of PE 36:2 in W. ginsengihum sample as an example, lipid subclass information was obtained from 141 Da of neutral loss of ethanolamine head group in positive ion MS/MS spectrum (Supplementary Fig. 10a

and 10c). Then, the structure could be proposed as PE 36:2 based on the accurate mass at m/z 744.5505 in positive ion mode. Accordingly, fatty acyl compositions were obtained as C18:1 due to the characteristic fragment ion at m/z 281.2 in negative ion mode (Supplementary Fig. 10b and 10c). Therefore, the preliminary structure was identified as PE 18:1_18:1. To obtain further information on C=C bonds, the lipid extract was subjected to the photocatalytic reaction and analyzed by LC-MS/MS. The EICs at m/z 744 before the reaction gave one predominant peak and two minor peaks at the right side for W. ginsengihum (Fig. 5b, c). The exact structures of these isomers could not be confirmed only based on the RTs. This difficulty was also encountered by other samples (Fig. 5b), and the isomeric peaks with different RTs were even observed for M. osloensis. However, by comparing with the EIC peaks at m/z 744 after the reaction, the peaks at 11.5 min could be easily identified as lipid possesses two Z configuration C=C bonds, and the peak at 11.8 min was Z/E isomers mixture. Notably, the peak at 12.1 min was not the isomer with two E-configuration C=C bonds, which may be misidentified if only based on the RT. For the identification of the location of C=C bonds, the [2 + 2] photocycloaddition product with MBF could be found with a 164 Da increase in m/z. The protonated diagnostic ions P1a and P2a were observed at m/z 493.4 and 625.4 (Fig. 5d and e), respectively, which indicated the location of C=C bond was Δ9 in each fatty acyl chain. Therefore, the final structures of the lipid at m/z 744 in P. ginsengihum sample could be identified as PE 18:1 (9Z)_18:1 (9Z) (predominant) and PE 18:1 (9Z)_18:1 (9E) (minor). The same results were obtained for samples P. syringae and P. citronellolis, whereas the right structure was PE 18:1 (11Z)_18:1 (11Z) for M. osloensis. Note that the sn-isomer could not be differentiated here. By using this approach, a total of 309 lipids were identified in these four bacterial samples (Supplementary Data), including FA, (27), phosphatidic acid (PA, 17), PC (49), PE (116), phosphatidylglycerol (PG, 53), DG (23), TG (7), ceramide (Cer, 5), and bis(-monoacylglycero)phosphate (BMP, 12, Supplementary Fig. 11). More importantly, there were 133 unsaturated lipids were identified both at the C=C geometric and location levels. Other representative analysis of lipid structures such as PE, PG, PC, PA, and DG were shown in Supplementary Fig. 12–16 in Supplementary Information. Note that it's not easy to directly align retention times associated with the underivatized lipids (that supposedly give stereochemistry) with that of the derivatized lipids which indicate the double bond position. Hence, we assign the underivatized positional isomers based on their relative contents, which may be compared by the peak intensities generated from fragments of their [2 + 2] cycloaddition products in CID. Interestingly, we did not find the coexistence of the positional isomers in the same bacterial sample.

In order to further validate the results for isomer identification obtained via the proposed method, the GC-MS analyze was conducted to examine the lipid structures in bacterial samples. Given that this method is suitable to analyze the FAs among lipids, the free FAs in the four kinds of samples were firstly methylated and then subjected to the GC-MS analysis. The location of C=C bonds in FAs were identified by both matching the reference spectral libraries and the retention times of lipid standards, whereas the configurations of C=C bonds in a specific lipid were confirmed solely by matching its retention times with the cis or trans standards (Supplementary Fig. 17–23). The identified FAs were listed in Supplementary Table 1, which shows that abundant trans FAs were detected in the bacterial samples, such as FA 17:1 (9E), FA 18:1 (11E), and FA 19:1 (10E), which were consistent with the observations in the LC-MS analysis.

This approach provides the opportunity to deeply investigate the landscape of lipid C=C bonds isomers in bacteria. Given the

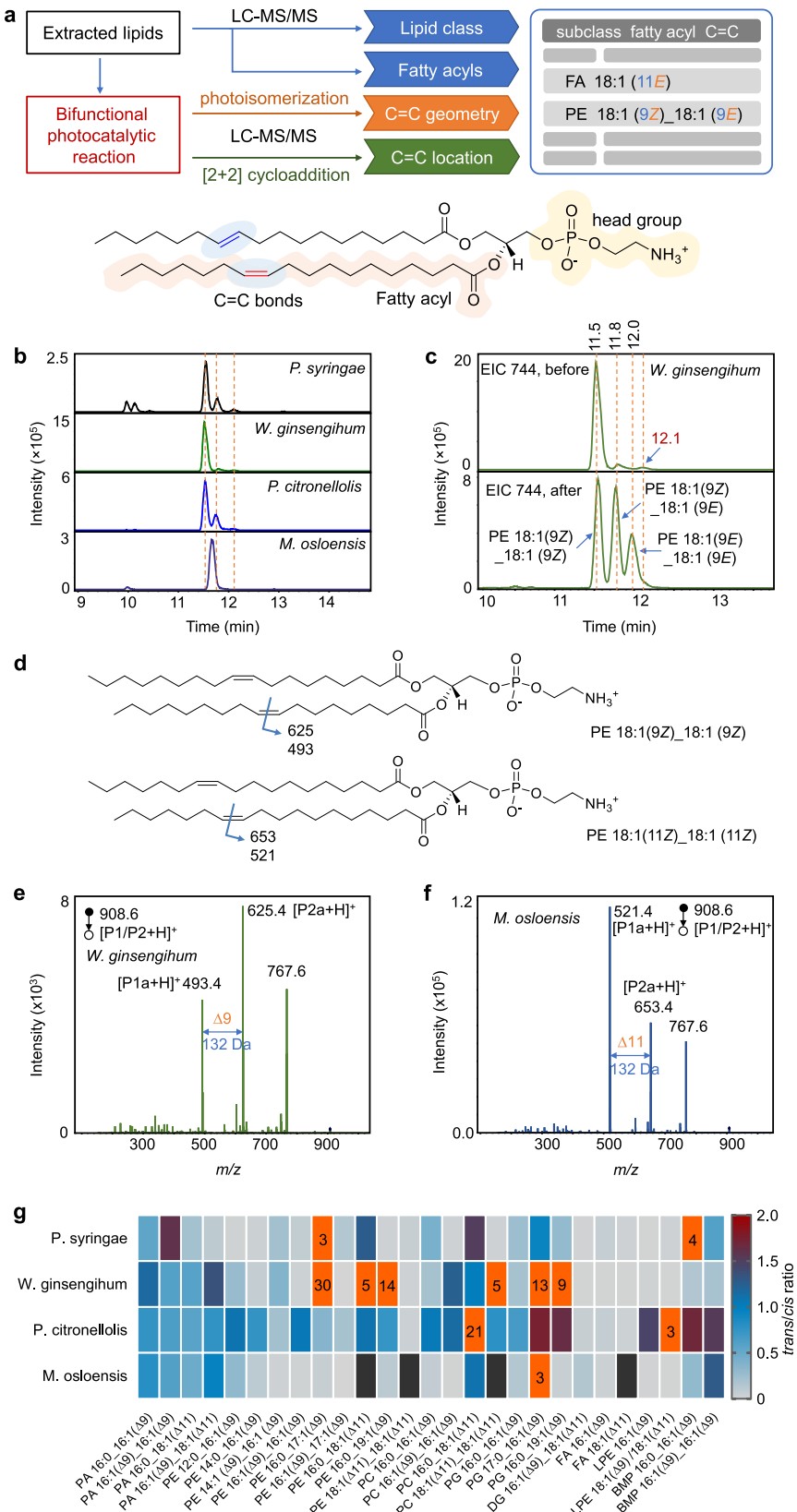

specific biological functions of *trans*-lipids in bacteria, we compared the ratios of *trans*-lipids to their *cis*-isomers for each lipid with defined C=C locations in the four bacterial samples. Figure 5g shows the heat map of the *trans*-to-*cis* ratios of each lipid (Supplementary Fig. 24), which revealed unique patterns of each bacterium. Different from the mammal samples, the bacteria

showed overall much more content of *trans* lipids, especially the phospholipids (Supplementary Table 2). For example, the *trans*-isomers of PAs are high in all four bacteria, the ratios are typically more than 50%. The ratios of *trans*-to-*cis*-isomers of PE 16:0_17:1(Δ9) and PE 16:0_19:1(Δ9) even reach 30 and 14 in *W. ginsengihum*. Nevertheless, the *trans* contents are very low in

**Fig. 5 Structural lipidomics analysis of bacterial samples derived from *Arabidopsis*. a** Workflow of the deep lipidmics by using the bifunctional photocatalytic reaction system and LC-MS/MS. PE 36:2 is used as a model to show the structure of lipids. **b** EICs of ions at *m/z* 744 which may correspond to PE 36:2 from four bacterial samples. **c** EICs of ions at *m/z* 744 from *W. ginsengihum* sample before and after the reaction. **d** Structures of PE 36:2 that show the positions of C=C bonds and the *m/z* values of the possible diagnostic ions of photocycloaddition products via CID. **e–f** MS/MS spectrum of the protonated photocycloaddition product (P1/P2) at *m/z* 908.6 from (**e**) *W. ginsengihum* and (**f**) *M. osloensis* samples, respectively. The peak at *m/z* 767.6 should correspond to the fragment ion by losing ethanolamine head group. **g** Heatmap that shows the EIC peak area ratios of lipids *trans*-isomers to their *cis*-counterpart from the four bacterial samples, except PE 18:1 (Δ11)_18:1 (Δ11) and PC 18:1 (Δ11)_18:1 (Δ11), which correspond to the ratios of PE 18:1 (11*Z*)_18:1 (11*E*) to PE 18:1 (11*Z*)_18:1 (11*Z*) and PC 18:1 (11*Z*)_18:1 (11*E*) to PC 18:1 (11*Z*)_18:1 (11*Z*), respectively. The black boxes denote the missing data. Boxes with orange color indicate the ratio values are >2, and the ratio values rounding to integers were labeled.

FA, DG, and LPE. For instance, the ratios of *trans*-isomers of FA 16:1 (Δ9) and FA 18:1 (Δ11) are only 1% in *W. ginsengihum*. Among the four bacteria, *P. citronellolis* contains more *trans* content of lipids (Supplementary Table 2). The *trans*-to-*cis* ratios are typically more than 20%, except for DG and FA (<10%). These results revealed the unique patterns of lipid compositions of bacteria at C=C isomeric levels, which indicates the different survival mechanisms of bacteria. These facts demonstrated the capability of the developed system for structural lipidomics analysis of complex biological samples.

**Track the changes of C=C bond positional isomers in brains of focal ischemia mice after reperfusion**. Ischemic stroke accounts for most cases of stroke, which may cause the death and disability of patients. However, the complicated pathophysiological mechanisms of cerebral ischemia still hindered us to prevent and cure strokes[44]. The rodent animal models of focal cerebral ischemia have been extremely useful in elucidating the patho-mechanism of human stroke[45]. The lipid changes have been observed in cerebral ischemia[45], nevertheless, the investigation of lipids metabolism at C=C isomer level is a challenge. Therefore, we first established the mouse middle cerebral artery occlusion (MCAO) model of focal ischemia, and identified the paired C=C positional isomers (Δ9 & Δ11, Δ11 & Δ13) of FA, PE, PC, and TG in mouse brain tissues. Then, analyzed the ratio changes of these isomers in normal and ischemic sides of the brain after different periods of reperfusion. We aim to examine the relationship between the relative changes of C=C positional isomers and the recovery of ischemia.

Unlike those in bacteria, the C=C bonds of lipids in the mouse brains were identified as Z configuration. The locations of C=C bonds could also be identified by the proposed workflow with the PCPI reaction system. Moreover, the peak area ratios of lipid positional isomers in normal (left) and ischemic (right) brains after 1- and 3-day reperfusion were compared, including FA 18:1 (9*Z*)/FA 18:1 (11*Z*), PE 16:0_18:1 (9*Z*)/PE 16:0_18:1 (11*Z*), PE 16:0_20:1 (11*Z*)/PE 16:0_20:1 (13*Z*), PC 18:0_18:1 (9*Z*)/PC 18:0_18:1 (11*Z*), PC 18:0_20:1 (11*Z*)/PC 18:0_20:1 (13*Z*), and TG 16:0_16:0_18:1 (9*Z*)/TG 16:0_16:0_18:1 (11*Z*) (Fig. 6). As shown in Fig. 6a, the relative concentration of FA 18:1 (9*Z*) is significantly higher (about three times) than FA 18:1 (11*Z*) isomer, and the higher content of Δ9-fatty acyl chain than its Δ11- isomer was also observed in PE, PC and TG (Fig. 6b, d, f). This result indicates that the biosynthesis of FA 18:1 (Δ9) is more productive than FA 18:1 (Δ11), as the FA 18:1 (Δ11) could only be biosynthesized from FA 16:1 (Δ9) via the elongation process in mammals. We also observed the ratio of FA 18:1 (9*Z*/11*Z*) in the right ischemic brain became significantly higher than that in its normal counterpart (*p* < 0.001), even after 1-day reperfusion. This change indicates that the ischemia has a distinct influence on the biosynthesis of the C=C positional isomers, which may be related to altered activities of elongases or desaturases. However, no significant difference was observed in the ratio of FA 18:1 (9*Z*/11*Z*) between left and right brains after 3 days of

reperfusion, which may provide a more complete recovery from the ischemia. Likewise, a significant difference in the ratio of PE 16:0_18:1 (9*Z*)/PE 16:0_18:1 (11*Z*) was also observed between normal and ischemic brains, but it has not recovered even after 3 days of reperfusion (Fig. 6b). Interestingly, PE 16:0_20:1 which contains a longer unsaturated fatty acyl chain shows more speedy restoration (Fig. 6c), i.e. the ratio of PE 16:0_20:1 (11*Z*/13*Z*) no longer has a significant difference between normal and ischemic brains after 3 days reperfusion. For PC 18:0_18:1 (9*Z*/11*Z*), PC 18:0_20:1 (11*Z*/13*Z*), and TG 16:0_16:0_18:1 (9*Z*/11*Z*), no significant differences were observed between normal and ischemic brains both after 1- and 3-day reperfusion. These results revealed that the biosynthesis of C=C positional isomers could be significantly influenced by brain ischemia and reperfusion, which indicates the metabolic reprogramming of unsaturated lipids in these processes. These findings further validated the applicability of the proposed method to a deeper understanding of the relationship between the structural specificities and the biological functions of lipids.

In summary, a visible-light-activated bifunctional photocatalytic reaction system with photoisomerization and [2 + 2] photocycloaddition was developed for the simultaneous identification of C=C bonds positional and geometric isomers of unsaturated lipids. Combined with LC-MS, this workflow enabled the comprehensive structural lipidomic analysis of bacterial samples, and revealed the unique patterns of lipid compositions of bacteria at the location and *cis*-*trans* isomeric levels of C=C bonds. In addition, this method allows the tracking of C=C positional isomers changes of brain ischemia in the mouse model, and provided further understanding of the lipid metabolism during the reperfusion process. This study not only offered a powerful tool for deep structural lipidomics analysis, but also provided insight into the triplet-energy-transfer-based photocatalytic reaction of carbonyls and C=C bonds.

## Methods

**Research compliance**. All animal experiments and procedures were approved by the committee of experimental animals of Tongji Medical College. All experimental protocols and animal handling procedures were performed in accordance with the National Institute of Health Guide for the Care and Use of Laboratory Animals (NIH Publications No. 80-23) revised 1996 and the experimental protocols were approved by the committee of experimental animals of Tongji Medical College.

**Visible-light-activated photocatalytic reaction**. A 40 W Blue LED Lamp (Kessil A160WE Tuna Blue) was used to trigger the photocatalytic reaction. Typically, 100 μL of reaction solution containing MBF (100 mM), olefins [lipid standards (5-10 mM) or lipid extract from biological samples], and photocatalyst Ir[dFppy]₂(dtbbpy)PF₆ (5 mol% of MBF) was prepared in methanol and acetonitrile (1:1, v/v) in a micro volume autosampler vial, and the dissolved oxygen was removed by bubbling nitrogen through a needle of a 2 mL syringe for 30 seconds. Then, the LED light was placed on the side of the glass vial to irradiate the reaction solution for 10 min. The distance between the lamp and the vial was ~8 cm. Two small fans were also used to maintain the reaction solution at room temperature. The solution was directly subjected to MS analysis after the reaction except for biological sampels, which was centrifuged at $16,000 \times g$ for 10 min before the MS ananlysis.

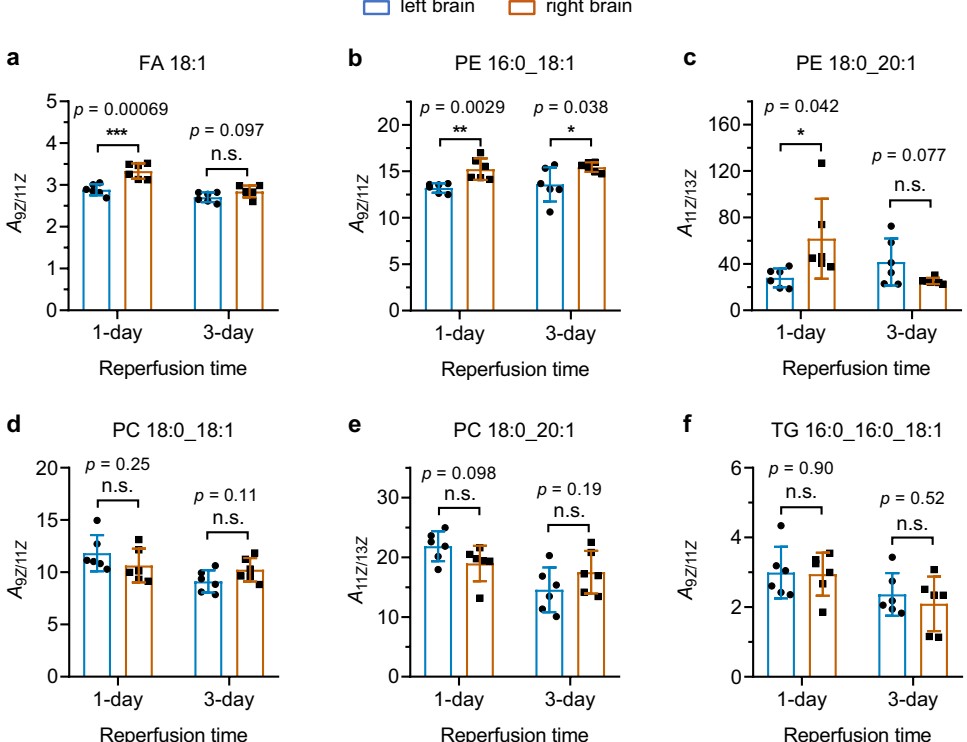

**Fig. 6 Comparison of the C=C locational isomer ratios between left and right brains of MCAO model mice after 1-day and 3-day reperfusion. a–f** Ratios of C=C locational isomers of (**a**) FA 18:1 FA 18:1 (9Z/11Z), **b** PE 16:0_18:1 (9Z /11Z), **c** PE 18:0_20:1 (11Z /13Z), **d** PC 18:0_18:1 (9Z/11Z), **e** PC 18:0_20:1 (11Z/13Z), and (**f**) TG 16:0_16:0_18:1(9Z /11Z) in left and right mouse brains. The C=C bonds were identified as Z configuration in these lipids using the proposed photoisomerization approach. The left brains were subjected to 1 h focal ischemia and then received reperfusion for 1 and 3 days. Two-sided Student's $t$ test was used without adjustment. $*p < 0.05$; $**p < 0.01$; $***p < 0.001$; n.s., no significant difference. Data are presented as mean values ±SD, $n = 6$ biologically independent replicates.

**Preparation of the bacterial samples**. Four bacterial samples were used in this study, including one model pathogen in *Arabidopsis* resistance study (*Pseudomonas syringae*, *P. syringae*) and three endophytic bacteria isolated from *Arabidopsis* leaves (*Weizmannia ginsengihum*,*W. ginsengihum*; *Pseudomonas citronellolis*,*P. citronellolis* and *Moraxella osloensis*,*M. osloensis*). To isolate the endophytic bacteria, *Arabidopsis thaliana* Col-0 plants were grown in unsterilized soil at 22 °C, with relative humidity of 60% and photoperiod at a 12:12 h light: dark cycle. 5 weeks later, mature leaves were detached and surface sterilized in 75% alcohol for 90 sec, followed by washing with sterile water three times. Leaves were then dried on sterile filter papers. Three leaves were put into a 1.5 mL Eppendorf tube filled with 500 μL 10 mM MgCl₂ and one steel ball. Leaves were ground with a grinder. After brief centrifugation, the supernatant was plated on R2A plates, and bacteria were grown at room temperature for 3 days. About 100 colonies were picked and cultured by R2A liquid medium on a shaker at 220 rpm and 28 °C overnight. 16 S rRNA fragments were amplified using primers 27 F (5'-AGAGTTTGATCCTGG CTCAG-3')/1492 R (5'-GGTTACCTTGTTACGACTT-3') and then sequenced. Sequence blast identified different endophytic bacteria and three of them were randomly selected for this study.

For the extraction of lipids from these samples, 400 μL Methyl tert-Butyl Ether (MTBE) was added into the 1.5 mL Eppendorf (EP) tube that contains bacterial cells collected by centrifugation, and subjected to vortex for 30 s and ultrasonic mixing for 10 min in ice water. Then, 80 μL MeOH and 200 μL H₂O were added successively with the same steps as above. After the vortex and sonicate, the mixture was centrifuged for 15 min at $660 \times g$ to separate phases. The upper MTBE phase was taken to a new EP tube and dried with nitrogen flow. The dried lipid extracts were redissolved in methanol and acetonitrile (1:1, v/v) and subjected to photocatalytic reaction.

**Middle cerebral artery occlusion (MCAO) model of focal ischemia**. Male C57BL/6 mice (wild-type) (8–10 weeks old, 23–25 g) were purchased from Wuhan University Center for Animal Experiment. The animals were maintained at temperature of 23 ± 1 °C with 50 ± 10% relative humidity in the center. Mice were allowed free access to food and water in a 12-h light/dark cycle. Focal cerebral ischemia was induced by transient occlusion of the right middle cerebral artery using a 6–0 silicone-coated nylon monofilament. The mice were firstly anesthetized with ketamine (100 mg/kg, i.p.) and xylazine (8 mg/kg, i.p.) before the surgery. Then, the right common carotid artery (CCA), the right external carotid artery

(ECA), and the right internal carotid artery (ICA) of the mice were isolated under the operating microscope, and a 6–0 suture was used to tie up the blood vessels at the origin of the ECA and at the distal end of the ECA, respectively, to temporarily occlude the right CCA and ICA. Subsequently, the nylon monofilament was introduced from a small incision into the ECA and was advanced to ICA at a position 9-10 mm distal from the carotid bifurcation to effectively block the middle cerebral artery (MCA). The acceptable diameter of the tip of coated filament is 180-220 μm. The filament remained inserted for 60 minutes, after which it was removed for reperfusion and the ECA was permanently ligated. During and after the surgery, the body temperature of mice was maintained at 37 ± 0.5 °C with a heating pad. The blood pressure (BP) was monitored through a femoral artery and the blood gas was analyzed at the end (i.e., 1 day and 3 days after tMCAO procedure). Subcutaneous normal saline (0.9%) was administered daily for adjusting the volume according to the animal's weight loss.

**Preparation of mouse brain tissues**. For the measurement of lipid changes, C57BL/6 mice were subject to 60 minutes ischemia followed by 1- and 3 days reperfusion. The brains of the mice were collected and cut into two halves from the middle line namely left (normal part) and right half (ischemic part). Then the cut parts were weighed and added into three times of methanol [weight (mg) to volume (mL)] for homogenization at 40 Hz for 2 min. The mixture was then diluted 30 times with methanol solution. The mixed methanol solution was divided into 300 μL fractions for further lipid extraction. Briefly, 300 μL of homogenate in methanol were added into 1 mL MTBE and vortexed for 10 s. Subsequently, the mixture was vibrated for 10 min. Then 300 μL of water was added and vortexed for 10 s to promote liquid-liquid stratification. After equilibration for 10 min, the mixture was centrifuged at $10,625 \times g$ for 10 min at 4 °C. A total of 900 μL of organic phase was transferred to a new tube for drying with nitrogen flow. The dried lipid extracts were reconstituted in 100 μL (equivalent of 10 mg original tissue samples) solution of methanol and acetonitrile (1:1, v/v), and subjected to photocatalytic reaction and LC-MS analysis.

**LC and MS analysis**. The data for reaction optimization and relative quantification of C=C location isomer were accomplished with LTQ-Orbitrap Elite mass spectrometer (Thermo Scientific, Germany). The main MS parameters were set up as follows: sheath gas (N₂): 40 arbitrary units; auxiliary gas (N₂): 5 units; capillary temperature: 350 °C; spray voltage: 3.8 kV; collision energy: 40 V; the resolution of

full mass scan: 30,000. In the experiments of MS/MS scan, the top 5 precursor ions (by intensity) were selected for MS/MS analysis, and the dynamic exclusion function was turned on for 10 s. The fragmentation of ions was induced by collision-induced dissociation, with a resolution of 15,000.

The analyses for C=C location and the lipids in bacterial samples were performed on a hybrid trapped ion mobility-quadrupole time-of-flight mass spectrometer (timsTOF Pro, Bruker Daltonics, Germany). These QToF instruments were operated to collect full scan MS data and MS/MS fragmentation spectra in the same analytical run with the parallel accumulation serial fragmentation method. Precursors for data-dependent acquisition were isolated within ±1 Th and fragmented with an ion mobility-dependent collision energy, which was linearly increased from 25 to 45 eV in positive mode, and from 35 to 55 eV in negative mode. The ion mobility was scanned from 0.6 to 1.95 Vs/cm$^2$. Low-abundance precursor ions with an intensity above a threshold of 100 counts but below a target value of 4000 counts were repeatedly scheduled and otherwise dynamically excluded for 0.2 min. TIMS ion charge control was set to 5E6. The TIMS dimension was calibrated linearly using four selected ions from the Agilent ESI LC/MS tuning mix. The ion source settings were: capillary voltage = 4.5 kV; endplate offset = 500 V; drying gas flow = 12.0 L/min; nebulizer gas = 5.0 bar; drying temperature = 250 °C. The data acquisition rate was set to 8 Hz over the mass range of $m/z$ 30–1000. The sodium adducts were used for fragmentation in CID to identify the C=C positional isomers of FAs, as both the deprotonated and protonated ions could not generate the anticipated diagnostic ions. For DG and TG, ammonium adducts were used for fragmentation to determine the isomers. Other glycerol phospholipids were fragmented in protonated forms to generate the diagnostic ions.

The LC-MS analysis was performed using a Waters ACQUITY UPLC BEH C18 column (2.1 mm × 100 mm, 1.7 mm) with the UltiMate 3000 UPLC system (DIONEX, Thermo Scientific, Germany). The temperature of the column was maintained at 40 °C for the separation of lipid species, and the flow rate of the mobile phase was set at 0.3 μL/min. The mobile phase for gradient elution consisted of mobile phase A [ACN-H$_2$O (60:40, v/v) mixed with 10 mM NH$_4$OAc and 0.1% formic acid] and mobile phase B [IPA-ACN (90:10, v/v) mixed with 10 mM NH$_4$OAc and 0.1% formic acid]. The optimal chromatographic gradient program for glycerol phospholipid standards and biological samples was set as follows: 30% B at 0~4 min, 30~52% B at 4~6 min, 52~63% B at 6~8 min, 63~68% B at 8~9 min, 68~74% B at 9~27 min, 74~80% B at 27~29 min, 80~99% B at 29~1 min, 99% B at 31~35 min; For FAs, the separation time was shortened and the optimal chromatographic gradient program is: 30% B at 0~4 min, 30~52% B at 4~6 min, 52~63% B at 6~8 min, 63~68% B at 8~10 min.

**Reporting summary**. Further information on research design is available in the Nature Research Reporting Summary linked to this article.

## Data availability

The MS data generated in this study have been deposited in ProteomeXchange Consortium (https://www.iprox.org/). Project ID: IPX0004110000. Source data are provided in this paper.

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

## Acknowledgements

This work was financially supported by the National Key Research and Development Program of China [2021YFC2700700 (S.M.C.)] and the National Natural Science Foundation of China [22074111 (S.M.C.), 22004092 (G.F.F), 22004093 (Q.Q.W.), 32070284 (G.Y.X.), 81571138 (Y.L.), and 82071485 (Y.L.)] and the Fundamental Research Funds for the Central Universities [WHU 2042021KF1020 (X.T.Q.)]. We also thank the support of the start-up funds of Wuhan University (S.M.C.) and the National Youth Talents Plan of China (S.M.C.). The numerical calculations have been done on the supercomputing system in the Supercomputing Center of Wuhan University. The GC-MS analysis was kindly supported by the lab of Professor Yi-Hung Chen at Wuhan University.

## Author contributions

S.M.C. and G.F.F. conceived and designed the experiments. G.F.F. performed most of the experiments. M.G. contributed to the reaction development. L.W.W. and X.T.Q. conducted the density functional theory calculations. J.Y.C. and Y.L. prepared the model of focal ischemia and collected the brain tissue samples. M.L.H. and G.Y.X. prepared the bacterial samples. G.F.F., S.M.C., and Q.Q.W. co-wrote the paper. All authors discussed the results and commented on the manuscript. S.M.C. supervised the overall research.

## Competing interests

The authors declare no competing interests.
