## [Peer Review File · Nature Communications]

REVIEWER COMMENTS

Reviewer #1 (Remarks to the Author):

The paper describes dual-resolving of positional and geometric isomerism of C=Cs by mass spectrometry via bifunctional photocycloaddition-photoisomerization reaction system. The paper could be published subject to several major revisions as indicated below:

1. Page 2, lines 39-41: "The reliable and panoptic resolving of C=C isomerism remains challenging, even though efforts have been made on the analysis of single isomeric type. Mass spectrometry (MS) has played important roles in the analysis of the C=C bonds isomerism due to its high sensitivity and structural elucidation capabilities"

The authors are not aware of the significant advantages of NMR spectroscopy in the field of C=C isomerism. Thus, Dais et al. [Anal. Methods 2015, 7, 5226–5238] utilized band selected 1H-13C gHSQC experiment to identify the terminal methyl protons of minor trans fatty acids in fish oil supplements which indicated a cross peak at $\delta = 13.72$ ppm. This low frequency chemical shift is characteristic of trans fatty acids [Fiori, L. et al., Food Chem. 2012, 134, 1088–1095]. The achievable resolution was found to be comparable to that of the 1D 13C-NMR spectrum in addition to much higher sensitivity due to indirect detection.

Furthermore, 2D 1H-13C HSQC-TOCSY experiment has been successfully employed to correlate carbon signals with the adjacent allylic protons of unsaturated fatty acids [Willker, W.; Leibfritz, D., Magn. Reson. Chem. 1998, 36, S79–S84; Vatele, J.-M. et al., Chem. Phys. Lipids 1998, 94, 239–250] and to analyze the positional distribution of unsaturated chains in TAG model compounds [Simova, S. et al., Chem. Phys. Lipids 2003, 126, 167–176].

Of particular interest is the application of gHSQC-TOCSY experiment for mixture analysis in DAG oils [Hatzakis, E. et al., J. Am. Oil. Chem. Soc. 2011, 88, 1695–1708]. Carbon - proton pairs connected over two bonds were observed for the allylic and bis-allylic carbons of the unsaturated chains which resulted in the unambiguous assignment of several carbons of the acyl chains of oleic, linoleic and linolenic acids. Furthermore, a semi-selective 1H-13C HSQC experiment was utilised [Willker, W. et al., J. Magn. Reson. 1997, 125, 216–219; Willker, W.; Leibfritz, D., Magn. Reson. Chem. 1998, 36, S79–S84] to obtain ultra-high resolution in the 13C double bond region (~0.7 Hz per point) and, thus, to assign and quantify individual unsaturated fatty acids, such as 18:1, 18:2 and 20:4 in body fluids (blood plasma and cerebrospinal fluid) and pig brain (grey and white matter) and polyunsaturated fatty acids in lipids of tissue and body fluids.

For a review see: Molecules 2017, 22, 1663.

2. With regard to computations:

(i) The use of the M06-2X/6-311+G(d,p)/SMD(acetonitrile)//M06-2X/6-31G(d) level of theory is correct since the M02X functional provides good thermochemical data.

(ii) Figure 2.

On the left it is written "radical addition" while on the right it is written "radical elimination". A correction, therefore, should be made. Furthermore, ΔG^\ddagger values are significantly different than those of ΔH^\ddagger . Please explain the importance of ΔS^\ddagger and, if possible, provide a molecular level interpretation.

(iii) Page 5, lines 109-110. "Moreover, the generated E-oleic acid 5 is 1.1 kcal/mol more stable than Z-oleic acid 1, which indicates the Z/E-alkene isomerization is kinetically and thermodynamically feasible." Since Z/E alkene isomerization is kinetically and thermodynamically feasible, it is not clear in a complex

mixture which contains Z/E alkene isomers and in addition polyunsaturated fatty acids how identification and quantification can be made.

(iv) Figure 4.

In the case of FA18:2 (9Z, 12Z), the bis-allylic CH₂ protons are very sensitive to hydrogen abstraction. I wonder whether the bifunctional photocycloaddition-photoisomerization reaction system could become very complicated in the case of complex polyunsaturated lipids.

Reviewer #2 (Remarks to the Author):

Feng et al. present some interesting photochemistry that facilitates (i) more efficient [2+2] cycloaddition of carbonyl compounds to olefins and (ii) evidence for cis-to-trans isomerisation. Experimental results support the role of energy transfer from the iridium-based photosensitizer to driving these processes and computational results show that these processes are energetically accessible and competitive on the diradical potential energy surface. The authors then exploit these photochemistries to identify the double bond position and geometry in unsaturated lipids ranging from simple fatty acids to complex glycerophospholipids. Application of the methods to biological extracts demonstrates some differences in lipid isomers between species or conditions. Overall, the chemistry is well articulated and mechanistic claims are well-supported by experimental and computational evidence. The bioanalytical applications are interesting but the advances over previous methodologies that exploit [2+2] Paterno-Buchi chemistries are not demonstrated or, in the case, of stereochemistry (cis/trans) unsupported by the data. In the absence of support for the novel claims and/or significant demonstration of incremental advance the manuscript is not suitable for publication in Nature Communications and should be recommended for a more specialised journal once the authors have addressed the specific issues raised below:

Page 1, Content: " The positional and geometric isomerism of C=C bonds are two of the essential attributes that determine the configurations of unsaturated lipids. "

Reviewer Comment: This is a confusing sentence. "Isomerism" is the process of converting one isomer to another. The configuration of double bonds encompasses the stereochemistry by definition; one does not determine the other they are the same thing.

Page 2, Content: "...stereoisomerism of C=C bonds in lipids includes positional and geometric (cis-trans or E-Z) isomerism"

Comment: More care needs to be taken with nomenclature. Here for example cis/trans are stereo- or geometric-isomers. db-position isomers are an example of regioisomers.

Page 2, Content: " C=C positional isomer shows "

Comment: Unclear what is intended here. Perhaps the authors are referring to isomer ratios?

Page 2, Content: "and lipids with cis-C=C in the fatty acyl chain were found to modulate plasma membrane domain registration/anti-registration.⁹"

Comment: This study refers to double bond position not stereochemistry. In eukaryotes double bonds are predominantly cis and this should be pointed out in the introduction.

Page 2, Content: "However, fatty acids with trans configurations are important components in the membrane lipids of various aerobic bacteria.¹⁴ "

Comment: Some discussion on whether there is relationship between aerobic and anaerobic organisms or environments would be of great interest here in the context of this study.

Page 2, Content: The reliable and panoptic resolving of C=C isomerism remains challenging, even though efforts have been made on the analysis of single isomeric type"

Comment: The "panoptic" term here and elsewhere is an overclaim. Surely a panoptic method would - by definition- resolve all isomers and give a complete structure elucidation. The authors themselves indicate that this is not the case here.

Page 2, Content: "Despite the progresses, these methods lack of the capability for simultaneous analysis the geometric configurations of C=C bonds. On the other hand, the accurate identification of E-Z isomer is even more difficult.^{6,26}"

Comment: The poor english makes it difficult to understand what is intended here. This is important because essentially this sets the gap in knowledge that is being addressed and the purported novel contribution. The assignment of double bond position by the PB and related chemistries has been exhaustively demonstrated and iteratively improved. At best this study is an example of the latter although even this claim would need to be supported by demonstrating that, for example, more isomers were resolved by the use of the photosensitizer than by direct activation of the carbonyl reagent. The claims around stereochemistry are not supported and should be downplayed or removed. The simple cis and trans isomers are well-resolved by chromatography so why would it be necessary to promote conversion of one to the other? Even for the complex lipids cis/trans isomers are resolved to an extent with the trans eluting (predictably) later than the cis on reversed phase. So what is the advantage of taking a biological extract that is potentially already a mixture of isomers and inducing isomerism? This will have a confounding effect noting that the isomerism is also reversible and the extent of the isomerism will be dependent on other aspects of lipid structure so cannot be used quantitatively as the authors claim.

Page 2, Content: "differential elution times"

Comment: The term "elution times" applies to chromatography and does not apply to IMS. Arrival time would be the equivalent terms for IMS.

Page 3, Content: " analysis"

Comment: Should be "analyse"

Page 3, Content: "we established an integrated workflow that enables the panoptic,"

Comment: Again, this is an overclaim. Panoptic would surely yield the full molecular structure and differentiate all types of isomers.

Page 5, Content: "decent"

Comment: Do the authors mean "efficient" here?

Page 5, Content: "chose"

Comment: Should be "chosen"

Page 5, Content: " Z-oleic acid [FA 18:1 (9Z)] "

Comment: oleic acid is Z by definition.

Page 8, Content: "d) Extracted ion chromatograms (EIC) of FA 18:1 (9Z) and FA 18:1 (9E) before and after the photocatalytic reaction in negative ion mode. "

Comment: As highlighted above, there is good and predictable separation of cis/trans isomers by LC. So if two features are detected with the same composition and double bond position the later eluting feature will be trans. The potential for isomerisation of cis to trans thus serves to confound the identification, i.e., how does one know what is nature trans versus the artefact? It is notable too that there is a small degree of reversibility of trans to cis. Given that the rate/efficiency of this photoconversion is not known for all isomers how can this be deployed quantitatively as suggested?

Page 9, Content: "Cis-isomers appeared always at a lower retention time than the trans-isomers, because the cis-fatty acyl moieties experience a weaker interaction with the alkyl chains of the reversed stationary phase due to the U-shaped geometry of the cis-isomers.³⁷ "

Comment: This is an unsatisfying explanation of the chromatographic behaviour of these lipids - surely this is more to do with the interaction of E/Z isomers with saturated C18 chains in HPLC stationary phase.

Page 9, Content: "original PC 16:0/18:1"

Comment: The m/z 760 is the (M+H)⁺ ion of of the PC indicated. This needs to be clarified.

Page 12, Content: "On the other hand, the quantification of the C=C geometric isomers could be simply based on the original LC peak areas of each isomer after identified their configurations. "

Comment: This is precisely my point. If the geometric isomers are resolved by chromatography what is the advantage, if any, of the cis-trans isomerisation.

Page 15, Content: "More importantly, there were 133 unsaturated lipids were identified both at the C=C geometric and location levels. "

Comment: In both biological studies there needs to be some independent verification of the structure(s) at least as it pertains to the key claims, i.e., are they cis or trans. Comparing to some GC-MS that would confirm the stereochemistry of at least of few of the unusual fatty acyl chains would be one way to do this.

Page 15, Content: "PE 16:0_17:1(Δ 9) and PE 16:0_19:1(Δ 9)"

Comment: These odd-chain species are likely to have chain branching. The potential for isomers arising from chain branching overlaid with shifts in retention time based on double bond position, stereochemistry, sn-position makes the assignment based on the changes in retention time with photoactivation speculative at best. In these complex samples the real weakness of the method is that

there is no ability to align retention times associated with the underivatized lipids (that supposedly give stereochemistry) with the retention times of the derivatized lipids that give the double bond position. This is a fundamental limitation that needs to be clearly outlined.

Reviewer #3 (Remarks to the Author):

This reports an innovative application of the Paternò-Büchi (PB) reaction by employing a photosensitizer and visible light to identify both double bond position and cis/trans geometry of lipids. The lipids undergo conversion of cis to trans double bonds via the addition and elimination of PB reagent. LCMS analysis of reaction products was employed to identify the original alkene stereochemistry. The authors also explored the reaction mechanism and showed proof-of-principle experiments on lipid standards to identify double bond geometry. The combination of the novel reaction and the application of the method to lipids in biological samples makes the paper a very strong one. There are some revisions recommended:

- 1) In some of the spectra, it looks like deprotonated, protonated, and sodium-cationized lipids are shown. It is not clear whether all of these ionized forms can be used to determine the isomers.
- 2) Ref 5 is from the same authors. How does the present study differ from Reference 5 and what was the new breakthrough in the present study that was not possible or not discovered in Reference 5? [Feng, G.; Hao, Y.; Wu, L.; Chen, S., A visible-light activated [2 + 2] cycloaddition reaction enables pinpointing carbon-carbon double bonds in lipids. *Chem. Sci.* 2020, 11, 7244-7251.] I am sure there is a difference between Ref 5 and the new study, and many readers would appreciate the clarity.

Other comments:

Figure 1: The overall LCMS workflow for identifying cis/trans isomers is not clear.

Figure 4b: m/z 187 should be labeled as [MBF+Na]⁺

Figure 4d: m/z 451, 504 and 625 do not appear to be correctly labeled based on the scale of the x-axis. For example, m/z 451 is far above the mid-point between 400 and 500. It cannot be 451. 625 is far too close to 600.

The gradient scale bar shown in Figure 5g does not show the whole range of colors. Nor is it clear what is being shown without a title for the gradient scale bar. It is a trans/cis ratio or something else? This should be added to the heat gradient scale bar.

Supporting Information: Mass accuracies should be tabulated for all identified fragments and added to the SI.

Response to Reviewers' Comments

Reviewer 1:

The paper describes dual-resolving of positional and geometric isomerism of C=Cs by mass spectrometry via bifunctional photocycloaddition-photoisomerization reaction system. The paper could be published subject to several major revisions as indicated below:

1. Page 2, lines 39-41: "The reliable and panoptic resolving of C=C isomerism remains challenging, even though efforts have been made on the analysis of single isomeric type. Mass spectrometry (MS) has played important roles in the analysis of the C=C bonds isomerism due to its high sensitivity and structural elucidation capabilities" The authors are not aware of the significant advantages of NMR spectroscopy in the field of C=C isomerism. Thus, Dais et al. [Anal. Methods 2015, 7, 5226–5238] utilized band selected 1H-13C gHSQC experiment to identify the terminal methyl protons of minor trans fatty acids in fish oil supplements which indicated a cross peak at $\delta = 13.72$ ppm. This low frequency chemical shift is characteristic of trans fatty acids [Fiori, L. et al., Food Chem. 2012, 134, 1088–1095]. The achievable resolution was found to be comparable to that of the 1D 13C-NMR spectrum in addition to much higher sensitivity due to indirect detection.

Furthermore, 2D 1H-13C HSQC-TOCSY experiment has been successfully employed to correlate carbon signals with the adjacent allylic protons of unsaturated fatty acids [Willker, W.; Leibfritz, D., Magn. Reson. Chem. 1998, 36, S79–S84; Vatele, J.-M. et al., Chem. Phys. Lipids 1998, 94, 239–250] and to analyze the positional distribution of unsaturated chains in TAG model compounds [Simova, S. et al., Chem. Phys. Lipids 2003, 126, 167–176].

Of particular interest is the application of gHSQC-TOCSY experiment for mixture analysis in DAG oils [Hatzakis, E. et al., J. Am. Oil. Chem. Soc. 2011, 88, 1695–1708]. Carbon - proton pairs connected over two bonds were observed for the allylic and bis-allylic carbons of the unsaturated chains which resulted in the unambiguous assignment of several carbons of the acyl chains of oleic, linoleic and linolenic acids.

Furthermore, a semi-selective 1H-13C HSQC experiment was utilised [Willker, W. et al., J. Magn. Reson. 1997, 125, 216–219; Willker, W.; Leibfritz, D., Magn. Reson. Chem. 1998, 36, S79–S84] to obtain ultra-high resolution in the 13C double bond region (~ 0.7 Hz per point) and, thus, to assign and quantify individual unsaturated fatty acids, such as 18:1, 18:2 and 20:4 in body fluids (blood plasma and cerebrospinal fluid) and pig brain (grey and white matter) and polyunsaturated fatty acids in lipids of tissue and body fluids.

For a review see: Molecules 2017, 22, 1663.

Reply: The authors appreciate the reviewer for summarizing the significant progress of NMR in the identification of lipid isomers. We have added the description for mentioning this point in the introduction section, and the related references were cited. Please see page 3 and references part in the manuscript.

2. With regard to computations:

(i) The use of the M06-2X/6-311+G(d,p)/SMD(acetonitrile)//M06-2X/6-31G(d) level of theory is correct since the M02X functional provides good thermochemical data.

Reply: Thanks for the recognition to our work.

(ii) Figure 2.

On the left it is written “radical addition” while on the right it is written “radical elimination”. A correction, therefore, should be made. Furthermore, ΔG^\ddagger values are significantly different than those of ΔH^\ddagger . Please explain the importance of ΔS^\ddagger and, if possible, provide a molecular level interpretation.

Reply: The “radical elimination” in Figure 2 has been changed to “ β -scission”. In this case, the “ β -scission” is used to describe the transformation from carbon-centered radicals **4/4a** to *E*-oleic acid **5** and oxygen-centered radical **2** via transition states **TS-3/4**. The corresponding description of this step is also revised. Please see page 6 in the manuscript.

In Figure 2, ΔG values are different from ΔH for these transition states and intermediates because of the entropy penalty. The entropy penalty corresponds to the value of ΔS . In this work, the entropy penalty initially occurs in the radical addition step as the combination of two molecules (**1** and **2**) only generate one molecular transition state (**TS-1/2**) or intermediate (**3/3a**). Considering that $\Delta G = \Delta H - T\Delta S$, the presence of entropy penalty renders the ΔG much smaller than that of ΔH . Therefore, their ΔG values are different from the ΔH values. This principle is also suitable for understanding the $\Delta G/\Delta H$ difference for intermediates **4/4a** and transition states **TS-3** and **TS-4**.

(iii) Page 5, lines 109-110. “Moreover, the generated *E*-oleic acid **5** is 1.1 kcal/mol more stable than *Z*-oleic acid **1**, which indicates the *Z/E*-alkene isomerization is kinetically and thermodynamically feasible.”

Since *Z/E* alkene isomerization is kinetically and thermodynamically feasible, it is not clear in a complex mixture which contains *Z/E* alkene isomers and in addition polyunsaturated fatty acids how identification and quantification can be made.

Reply: We understand the concern of the reviewer. In fact, we have a complete workflow to qualify and quantify the C=C isomers of unsaturated lipids and their mixtures. The *Z/E* alkene isomerization was used to identify the configurations of the C=C bonds, but the quantification of the unsaturated lipids was based on their individual peak areas in liquid chromatography (LC)-mass spectrometry (MS) before the photochemical reactions. So, the *Z/E* alkene isomerization will not affect the accuracy of the quantification. The LC separation could also reduce the complexity of the mixture.

Take the analysis of phosphatidylethanolamine PE 36:2 in bacterial sample as an example, the qualification and quantification could be performed, respectively (Figure R1 below). For the qualification, the lipids extract was divided into two parts, one for conventional LC-MS/MS analysis to obtain the information of lipid class and the compositions of fatty acyl chains (e.g. PE 18:1_18:1). The other part was firstly subjected to the bifunctional photochemical reaction and then analyzed by LC-MS/MS (Figure R1a). The reaction products could be separated and eluted by LC and analyzed by MS or MS/MS. For PE 36:2, the protonated [P1/P2+H]⁺ at *m/z* 908.6 were fragmented by collision induced dissociation, and the diagnostic ions at *m/z* 521.4 and 653.4 could be produced to indicate the C=C position at $\Delta 9$ in both fatty acyl chains (Figure R1b). Thus, the PE 36:2 could be identified as PE 18:1 ($\Delta 9$)_18:1 ($\Delta 9$). For the identification of C=C configuration, the extracted ion chromatograms (EIC) of PE 36:2 at *m/z* 744 before and after the photocatalytic reaction were obtained and compared (Figure R1c). Before the reaction, one predominant peak at retention time (RT) 11.5 min was observed, accompanied by two minor peaks at RT 11.8 and 12.1 min. They were speculated as the *cis/trans* isomers of PE 18:1 ($\Delta 9$)_18:1 ($\Delta 9$). After the photoisomerization reaction, the relative intensity of the peak at RT 11.8 min increased greatly, and a new peak at RT 12.0 min appeared. This result indicates the formation of the photoisomerized products with one and two *trans* C=C bonds. So, we could identify the original peaks at RT 11.5 and 11.8 min as PE 18:1 (9*Z*)_18:1 (9*Z*) and PE 18:1 (9*Z*)_18:1 (9*E*), respectively. This result could also help to clarify that the peak at RT 12.1 min was not the geometric isomer of PE 18:1 ($\Delta 9$)_18:1 ($\Delta 9$) because of the mismatched

retention time. To quantify these isomers in the bacterial sample, the individual peak area in the EIC before the bifunctional reaction will be used (the top panel of Figure R1c).

Figure R1. Structural lipidomics analysis of bacterial samples. (a) Workflow of the deep lipidomics by using the bifunctional photocatalytic reaction system and LC-MS/MS. PE 36:2 is used as a model to show the structure of lipids. (b) MS/MS spectrum of the protonated photocycloaddition product (P1/P2) of PE 36:2 at m/z 908.6 from bacterial sample. Peak at m/z 767.6 corresponds to the fragment ion by losing ethanolamine head group. (c) EICs of ions at m/z 744 for protonated PE 36:2 before and after the reaction.

(iv) Figure 4.

In the case of FA18:2 (9Z, 12Z), the bis-allylic CH₂ protons are very sensitive to hydrogen abstraction. I wonder whether the bifunctional photocycloaddition-photoisomerization reaction system could become very complicated in the case of complex polyunsaturated lipids.

Reply: Thanks for this comment. We agree with the reviewer's opinion that the bis-allylic CH₂ protons in FA 18:2 (9Z, 12Z) are very reactive for hydrogen atom abstraction. One may surmise that the hydrogen atom abstraction pathway might compete with the radical addition to the alkene, which ultimately results in a complicated outcome for the identification of C=C bonds location and E-Z configuration. DFT calculations were then performed to study these two competitive pathways using FA18:2 (9Z, 12Z) as the reactant (Figure R2 below). The computational

results support that the radical addition to alkene *via* TS-5 is kinetically favored over the hydrogen atom abstraction *via* TS-6. Therefore, the hydrogen atom abstraction pathway cannot outcompete the radical addition step. Moreover, our experimental studies also suggest that the analysis of complex polyunsaturated lipids are well compatible in this work (see Figure 4 in manuscript and Supplementary Datasheet). This complementary computational result was also added into the manuscript (page 15) and Supplementary Information (Figure S4).

Figure R2. Computational study of the comparison between hydrogen atom abstraction and radical addition to alkene in the case of FA18:2 (9Z, 12Z).

Reviewer 2:

1. Feng et al. present some interesting photochemistry that facilitates (i) more efficient [2+2] cycloaddition of carbonyl compounds to olefins and (ii) evidence for *cis*-to-*trans* isomerisation. Experimental results support the role of energy transfer from the iridium-based photosensitizer to driving these processes and computational results show that these processes are energetically accessible and competitive on the diradical potential energy surface. The authors then exploit these photochemistries to identify the double bond position and geometry in unsaturated lipids ranging from simple fatty acids to complex glycerophospholipids. Application of the methods to biological extracts demonstrates some differences in lipid isomers between species or conditions. Overall, the chemistry is well articulated and mechanistic claims are well-supported by experimental and computational evidence. The bioanalytical applications are interesting but the advances over previous methodologies that exploit [2+2] Paterno-Büchi chemistries are not demonstrated or, in the case, of stereochemistry (*cis/trans*) unsupported by the data. In the absence of support for the novel claims and/or significant demonstration of incremental advance the manuscript is not suitable for publication in Nature Communications and should be recommended for a more specialised journal once the authors have addressed the specific issues raised below.

Reply: We thank the reviewer for the recognition valuable suggestions to our work. Compare with the previous studies that exploit [2+2] Paternò-Büchi reaction to identification the location of C=C in lipids, this work developed a bifunctional photocatalytic reaction system that could enable the identification of both the location and *cis-trans* configurations of C=C bonds in unsaturated lipids. This reaction has two different pathways: [2+2]

photocycloaddition of carbonyl group with C=C bond, and *cis-trans* photoisomerization of C=C bond. The photocycloaddition reaction could be used to identify the location of C=C bonds with tandem mass spectrometry (MS), whereas the photoisomerization reaction could tell the configurations of C=C bonds when combined with LC-MS.

For the conventional LC-MS analysis, there is no good way to distinguish the configuration of C=C bond in lipids if only separate *cis* or *trans* isomer is available. This is the case in many systems. For example, the unsaturated lipids in eukaryotes are predominantly *cis* type, but the *trans* structure may also be the predominant configuration of lipids in many aerobic bacteria. Especially, we may not predict which configuration of C=C bonds for a specific lipid is predominant in bacterial samples. In addition, many natural products are only *cis* or *trans* in different kinds of samples. So, it's hard to identify the configurations of C=C bonds only based on the retention times in LC. By matching their elution times with that of the known standards, the structures of some of the lipids may be identified. However, the acquirement of standards for numerous lipids in complicated biological systems is quite difficult. Furthermore, the varied LC conditions may also confuse the assignment of the isomers. Although LC may separate the *cis*- and *trans*- isomers if both *cis-trans* isomers exist with appropriate amount in the sample, the possibility of the existence of other forms of isomers (e.g., lipid isomers with branched chains) in complicated biological samples may mislead the identification of the peaks in LC. So, the *cis-trans* configurations of C=C bonds were not assigned in a typical lipidomic study. This drawback hinders the deep understanding of the biological functions of *cis-trans* isomers of lipids. Therefore, the additional dimensional information is necessary to assist the identification of the *cis-trans* configurations of C=C bonds.

In this study, we utilize the characteristics of differential photoisomerization of *cis/trans* lipid isomers and their LC patterns to distinguish them. Take the analysis of *cis-trans* isomers of FA 18:1 as an example, the configurations of C=C bonds could be accurately identified by comparing their LC patterns before and after the photoisomerization reaction. As shown in Figure R3 below, both the FA 18:1 (9Z) and FA 18:1 (9E) has a single peak before the reaction, and the configurations of C=C bonds are difficult to be identified if there are no lipid standards in hand. However, we could easily assign it as *Z*-configuration if a new peak corresponds to its *E*-isomer appears on the right side of the original peak in the extracted ion chromatogram (EIC) after the photoisomerization reaction. In contrast, the C=C could be identified as *E*-configuration if there is a new minor peak corresponds to its *Z*-isomer appear on the left of the original peak in the extracted ion chromatogram after the reaction. Based on this strategy, the configurations of C=C bonds in polyunsaturated lipids could also be identified (see Figure 4 in the manuscript). Furthermore, this method could also help to exclude some isomers that might be mistaken for *cis-trans* ones in the analysis of lipids in complicated biological samples. For instance, the EICs of *m/z* 744 corresponds to PE 36:2 from bacterial samples show multiple isomers (Figure R4a below). Take the EIC from *W. ginsengihum* sample as an example (Figure R4b), a predominant peak at retention time (RT) 11.5 min and two minor peaks at RT 11.8 and 12.1 min were observed. The analytical results of lipid class, fatty acyl chain, and C=C location indicate the structure of PE 18:1 (Δ^9)_{18:1} (Δ^9), but the configurations of C=C bonds are unknown. One may speculate that these three peaks are PE 18:1 (9Z)_{18:1} (9Z), PE 18:1 (9Z)_{18:1} (9E) and PE 18:1 (9E)_{18:1} (9E), respectively, based on the different retention times. However, the results of photoisomerization reaction indicate that the retention time of PE 18:1 (9E)_{18:1} (9E) should be 12.0 min. The peak at RT 12.1 min may be ascribed from another form of isomer. After the identification of the lipid structures, the quantification of PE 18:1 (9Z)_{18:1} (9Z) and PE 18:1 (9Z)_{18:1} (9E) in bacterial sample could be performed base on their peak areas in EIC before the bifunctional photocatalytic reaction.

Overall, we have developed a bifunctional photocatalytic reaction system that could simultaneously resolve the C=C location and *E-Z* isomers of lipids. Compare with the previous studies using [2+2] Paternò-Büchi reaction that

only solved the problem of the location identification, the novel strategy utilizing additional photoisomerization reaction could also provide more reliable identification of C=C configurations. We believe this strategy is a significant advance in the deep lipidomics analysis, and would have important applications in revealing the biological functions of *cis-trans* lipid isomers.

Figure R3. Extracted ion chromatograms (EICs) of (a) FA 18:1 (9Z) and (b) FA 18:1 (9E) before and after the photocatalytic reaction in negative ion mode.

Figure R4. (a) EICs of ions at m/z 744 which may correspond to PE 36:2 from four bacterial samples. (b) EICs of ions at m/z 744 from *W. ginsengihum* sample before and after the photocatalytic reaction.

2. Page 1, Content: "The positional and geometric isomerism of C=C bonds are two of the essential attributes that determine the configurations of unsaturated lipids."

Reviewer Comment: This is a confusing sentence. "Isomerism" is the process of converting one isomer to another. The configuration of double bonds encompasses the stereochemistry by definition; one does not determine the other they are the same thing.

Reply: Thanks for pointing out this mistake. This sentence was modified into "The position and configuration of C=C bonds are two of the essential attributes that determine the structures of unsaturated lipids."

3. Page 2, Content: "...stereoisomerism of C=C bonds in lipids includes positional and geometric (*cis-trans* or E-Z) isomerism"

Comment: More care needs to be taken with nomenclature. Here for example cis/trans are stereo- or geometric-isomers. db-position isomers are an example of regioisomers.

Reply: The sentence was changed into "The types of isomers of C=C bonds in lipids includes positional and geometric (*cis-trans* or *E-Z*) isomers"

4. Page 2, Content: "C=C positional isomer shows"

Comment: Unclear what is intended here. Perhaps the authors are referring to isomer ratios?

Reply: It should be "C=C positional isomer ratio...". We have modified in the manuscript.

5. Page 2, Content: "and lipids with cis-C=C in the fatty acyl chain were found to modulate plasma membrane domain registration/anti-registration.⁹"

Comment: This study refers to double bond position not stereochemistry. In eukaryotes double bonds are predominantly cis and this should be pointed out in the introduction.

Reply: We have modified the description and added the information about eukaryotes double bonds in the introduction.

6. Page 2, Content: "However, fatty acids with trans configurations are important components in the membrane lipids of various aerobic bacteria.¹⁴"

Comment: Some discussion on whether there is relationship between aerobic and anaerobic organisms or environments would be of great interest here in the context of this study.

Reply: More discussion about the fatty acids in aerobic and anaerobic bacteria were added in the introduction.

7. Page 2, Content: The reliable and panoptic resolving of C=C isomerism remains challenging, even though efforts have been made on the analysis of single isomeric type"

Comment: The "panoptic" term here and elsewhere is an overclaim. Surely a panoptic method would -by definition- resolve all isomers and give a complete structure elucidation. The authors themselves indicate that this is not the case here.

Reply: We thank the reviewer for this pertinent suggestion. The word "panoptic" was deleted or modified through the paper.

8. Page 2, Content: "Despite the progresses, these methods lack of the capability for simultaneous analysis the geometric configurations of C=C bonds. On the other hand, the accurate identification of E-Z isomer is even more difficult.^{6,26}"

Comment: The poor English makes it difficult to understand what is intended here. This is important because essentially this sets the gap in knowledge that is being addressed and the purported novel contribution. The assignment of double bond position by the PB and related chemistries has been exhaustively demonstrated and iteratively improved. At best this study is an example of the latter although even this claim would need to be supported by demonstrating that, for example, more isomers were resolved by the use of the photosensitizer than by direct activation of the carbonyl reagent. The claims around stereochemistry are not supported and should be downplayed or removed. The simple cis and trans isomers are well-resolved by chromatography so why would it be necessary to promote conversion of one to the other? Even for the complex lipids cis/trans isomers are resolved to an extent with the trans eluting (predictably) later than the cis on reversed phase. So what is the

advantage of taking a biological extract that is potentially already a mixture of isomers and inducing isomerism? This will have a confounding effect noting that the isomerism is also reversible and the extent of the isomerism will be dependent on other aspects of lipid structure so cannot be used quantitatively as the authors claim.

Reply: We apologize for the confusion in this section. We mean that the previous methods lack of the capability for simultaneous identification of the positional and geometric configurations of C=C bonds. As we demonstrated in Question 1, the PB and related reactions were typically used to resolve the position of C=C bonds in unsaturated lipids, but the bifunctional photocatalytic reaction system developed in this study could be employed to simultaneously identified the position and configuration of C=C bonds.

Although LC may separate the *cis*- and *trans*- isomers if both *cis-trans* isomers exist with appropriate amount in the sample, the possibility of the existence of other forms of isomers (e.g., lipid isomers with branched chains or different C=C positions) in complicated biological samples may mislead the identification of the peaks in LC. The greater difficulty is to confirm the lipid C=C configuration of a single peak in LC when only one kind of the *cis/trans* isomer is available. In this case, no reference information of a counterpart isomer is accessible. This is the case in many systems. For example, the unsaturated lipids in eukaryotes are predominantly *cis*, but the *trans* structure may also be the predominant configuration of lipids in many aerobic bacteria. In addition, many natural products are only *cis* or *trans* in different kinds of samples. So, it's hard to identify the configurations of C=C bonds only based on the retention times in LC. By matching their elution times with that of the known standards, the structures of some of the lipids may be identified. However, the acquirement of standards for numerous lipids in complicated biological systems is quite difficult. Furthermore, the varied LC conditions may also confuse the assignment of the isomers. So, the *cis-trans* configurations of C=C bonds were not assigned in a typical lipidomic study. This drawback hinders the deep understanding of the biological functions of *cis-trans* isomers of lipids. Therefore, the additional dimensional information is necessary to assist the identification of the *cis-trans* configurations of C=C bonds.

In this study, we utilize the characteristics of distinctive photoisomerization of *cis/trans* lipid isomers and their LC pattern to distinguish them. Take the analysis of *cis-trans* isomers of FA 18:1 in Question 1 as an example, the configurations of C=C bonds could be accurately identified by comparing their LC patterns before and after the photoisomerization reaction (as shown in Figure R3 above). These different isomerization and LC separation characteristics obtained from the photoisomerization reaction could help to reliably identify the configuration of C=C bond in lipids. For the identification of the C=C location in unsaturated lipids with the proposed visible-light-activated photocatalytic reaction, we found that more unsaturated phospholipids could be identified with defined C=C locations (please see Supplementary Datasheet for the identified lipids in bacteria), such as PE, PA and PG, when compared with conventional UV-light-activated PB reaction. These results indicated the features of the developed method.

On the other hand, the *Z/E* alkene isomerization was only used to identify the configurations of the C=C bonds, but the quantification of the unsaturated lipids was based on their individual peak areas in LC-MS before the photochemical reactions. So, the *Z/E* alkene isomerization will not affect the accuracy of the quantification.

9. Page 2, Content: "differential elution times"

Comment: The term "elution times" applies to chromatography and does not apply to IMS. Arrival time would be the equivalent terms for IMS.

Reply: Thanks for pointing out this mistake. The "arrival times" has been added for the IMS method.

10. Page 3, Content: " analysis"

Comment: Should be "analyse"

Reply: They have been corrected.

11. Page 3, Content: "we established an integrated workflow that enables the panoptic,"

Comment: Again, this is an overclaim. Panoptic would surely yield the full molecular structure and differentiate all types of isomers.

Reply: The word "panoptic" was modified.

12. Page 5, Content: "decent"

Comment: Do the authors mean "efficient" here?

Reply: Yes, the word "decent" has been changed into "efficient".

13. Page 5, Content: "chose"

Comment: Should be "chosen"

Reply: The word "chose" has been changed into "chosen".

14. Page 5, Content: " Z-oleic acid [FA 18:1 (9Z)] "

Comment: oleic acid is Z by definition.

Reply: The "Z-oleic acid" has been changed into "oleic acid".

15. Page 8, Content: "d) Extracted ion chromatograms (EIC) of FA 18:1 (9Z) and FA 18:1 (9E) before and after the photocatalytic reaction in negative ion mode. "

Comment: As highlighted above, there is good and predictable separation of *cis/trans* isomers by LC. So if two features are detected with the same composition and double bond position the later eluting feature will be *trans*. The potential for isomerisation of *cis* to *trans* thus serves to confound the identification, i.e., how does one know what is nature *trans* versus the artefact? It is notable too that there is a small degree of reversibility of *trans* to *cis*. Given that the rate/efficiency of this photoconversion is not known for all isomers how can this be deployed quantitatively as suggested?

Reply: As mentioned above, although LC may separate the *cis-* and *trans-* isomers if both *cis-trans* isomers exist with appropriate amount in the sample, the possibility of the existence of other forms of isomers in complicated biological samples may mislead the identification of the peaks in LC (Figure R5a and R5b below). Even more difficult is to confirm the lipid C=C configuration of a single peak in LC when only one kind of the *cis/trans* isomer is present. In this case, no reference information of a counterpart isomer is available.

In the analysis of the *cis/trans* isomers mixture (e.g., FA 18:1 (9Z) and FA 18:1 (9E)) by the proposed photoisomerization reaction, the peak intensity/area ratio of FA 18:1 (9Z) to FA 18:1 (9E) in EICs before and after the reaction will decrease due to the preferential conversion of Z- to E- configurations (Figure R5c and R5d below). Although the exact rate/efficiency of this photoconversion is not known for all isomers, the preferred *cis-to-trans* photoisomerization of unsaturated lipids in this photocatalytic reaction system was confirmed by both the computational and experimental results. Therefore, it could quantitatively indicate the preference of the

conversion, by comparing the change of peak intensity/area ratio of the *cis/trans* isomers in EICs before and after the reaction. Then, we could assign the *cis/trans* configurations to the two isomers.

Figure R5. Simulated extracted ion chromatograms (EICs) of FA 18:1 (9Z) and FA 18:1 (9E) before and after the photocatalytic reaction.

16. Page 9, Content: "Cis-isomers appeared always at a lower retention time than the trans-isomers, because the cis-fatty acyl moieties experience a weaker interaction with the alkyl chains of the reversed stationary phase due to the U-shaped geometry of the cis-isomers."

Comment: This is an unsatisfying explanation of the chromatographic behaviour of these lipids - surely this is more to do with the interaction of E/Z isomers with saturated C18 chains in HPLC stationary phase.

Reply: The explanation has been modified as the reviewer's suggestion. Please see page 9 in the manuscript.

17. Page 9, Content: "original PC 16:0/18:1"

Comment: The m/z 760 is the (M+H)⁺ ion of of the PC indicated. This needs to be clarified.

Reply: The indication of [M+H]⁺ has been added at the end of the sentence.

18. Page 12, Content: "On the other hand, the quantification of the C=C geometric isomers could be simply based on the original LC peak areas of each isomer after identified their configurations. "

Comment: This is precisely my point. If the geometric isomers are resolved by chromatography what is the advantage, if any, of the cis-trans isomerisation.

Reply: We understand the concern of the reviewer. However, although the geometric isomers of C=C bonds may be separated by LC, one might not be that confident to confirm the precise configurations, especially for the complicated biological systems that may contain unexpected forms of isomers. In this study, by comparing the chromatographic behaviors before and after the photoisomerization reaction of unsaturated lipids, we could more reliably identify the original configurations of their C=C bonds. More discussions could be found in the replies of Questions 1, 8 and 15.

19. Page 15, Content: "More importantly, there were 133 unsaturated lipids were identified both at the C=C geometric and location levels. "

Comment: In both biological studies there needs to be some independent verification of the structure(s) at least as it pertains to the key claims, i.e., are they *cis* or *trans*. Comparing to some GC-MS that would confirm the stereochemistry of at least of few of the unusual fatty acyl chains would be one way to do this.

Reply: We thank the suggestions of the reviewer. Comparing with the conventional lipidomics methods, the feature of the proposed strategy is to identify the configurations of the C=C bonds in unsaturated lipids. In order to further validate the results obtained *via* the proposed method, the GC-MS analysis was conducted to examine the lipid structures. Given that the GC-MS method is suitable to analyze the fatty acids among lipids, the detailed analyses were performed for the four kinds of bacterial samples as those samples contain many *cis* or *trans* fatty acids. The free fatty acids were firstly methylated and then subjected to the GC-MS analysis. The location of C=C bonds in fatty acids were identified by both matching the standard spectral libraries and the retention times of lipid standards, whereas the configurations of C=C bonds in a specific lipid was confirmed solely by matching its retention times with the *cis* or *trans* standards. The results show that abundant *trans* fatty acids were detected in the bacterial samples, such as FA 17:1 (9*E*), FA 18:1 (11*E*), and FA 19:1 (10*E*), which were consistent with the observations in the LC-MS analysis. For FA 17:1 and FA 19:1 with odd-chains, the results indicated the predominantly straight-chain structures. The results were shown in Figure S17-S23 and Table S1 in the Supplementary Information, and the description of GC-MS analysis was added in lines 330-338 on page 16 and 17 in the manuscript.

20. Page 15, Content: "PE 16:0_17:1(Δ 9) and PE 16:0_19:1(Δ 9)"

Comment: These odd-chain species are likely to have chain branching. The potential for isomers arising from chain branching overlaid with shifts in retention time based on double bond position, stereochemistry, *sn*-position makes the assignment based on the changes in retention time with photoactivation speculative at best. In these complex samples the real weakness of the method is that there is no ability to align retention times associated with the underivatized lipids (that supposedly give stereochemistry) with the retention times of the derivatized lipids that give the double bond position. This is a fundamental limitation that needs to be clearly outlined.

Reply: We agree with the reviewer that different forms of isomers may complicate the assignment of the lipid structures. Although LC may alleviate this issue by optimizing the separation conditions (e.g. gradient, mobile phase, column), this is still a common challenge in the LC-MS-based lipidomics. However, the photochemical reaction of unsaturated lipids might provide additional information for identifying these isomers, by observing the new appeared peaks, shifts of LC retention time or changes of peak intensity ratio before and after the reactions. For example, if the retention time of a chain branching isomer overlaid exactly with that of the *trans* isomer of a *cis* lipid, we might still identify this *cis* lipid by comparing the change of intensity ratio of these two peaks. But we admit that some extreme conditions might exist if multiple isomers overlaid with each other. In these cases, the reliable identification of these isomers may be difficult.

We also agree that it is not easy to directly align retention times associated with the underivatized lipids (that supposedly give stereochemistry) with the retention times of the derivatized lipids that give the double bond position. This weakness might compromise the simultaneous identification of location and configuration of C=C bond in lipids in some complicated biological samples. However, in some cases, we may still assign the underivatized positional isomers of lipids based on their relative contents. For example, the relative contents of FA 18:1 (9*Z*) and FA 18:1 (11*E*) may be compared based on the fragment peak intensities of their [2+2] cycloaddition products in collision induced dissociation. Then, the peaks of underivatized FA 18:1 (9*Z*) and FA 18:1

(11E) in EIC would be assigned by comparing their relative intensities. For the bacterial samples, interestingly, we did not find the coexistence of the positional isomers in the same bacterial sample. This limitation may not influence the identification of the lipid isomers in these samples. To mention the possible limitation, we added a description in the revised manuscript. Please see page 16 in the manuscript.

Reviewer 3:

1. This reports an innovative application of the Paternò–Büchi (PB) reaction by employing a photosensitizer and visible light to identify both double bond position and cis/trans geometry of lipids. The lipids undergo conversion of cis to trans double bonds via the addition and elimination of PB reagent. LCMS analysis of reaction products was employed to identify the original alkene stereochemistry. The authors also explored the reaction mechanism and showed proof-of-principal experiments on lipid standards to identify double bond geometry. The combination of the novel reaction and the application of the method to lipids in biological samples makes the paper a very strong one.

Reply: We thank the recognition of the reviewer to our work.

2. In some of the spectra, it looks like deprotonated, protonated, and sodium-cationized lipids are shown. It is not clear whether all of these ionized forms can be used to determine the isomers.

Reply: The sodium adducts were used for fragmentation in collision induced dissociation to identify the C=C positional isomers of fatty acids, as both the deprotonated and protonated ions could not generate the anticipated diagnostic ions. For DG and TG, ammonium adducts were used for fragmentation to determine the isomers. Other glycerol phospholipids were fragmented in protonated forms to generate the diagnostic ions. This description was added to the Supplementary Information in the “Liquid chromatographic and mass spectrometric analysis” part of Methods section.

3. Ref 5 is from the same authors. How does the present study differ from Reference 5 and what was the new breakthrough in the present study that was not possible or not discovered in Reference 5? [Feng, G.; Hao, Y.; Wu, L.; Chen, S., A visible-light activated [2 + 2] cycloaddition reaction enables pinpointing carbon–carbon double bonds in lipids. *Chem. Sci.* 2020, 11, 7244-7251.] I am sure there is a difference between Ref 5 and the new study, and many readers would appreciate the clarity.

Reply: In our previous study (*Chem. Sci.* 2020, 11, 7244-7251), we developed a visible-light activated [2+2] cycloaddition reaction of anthraquinone with unsaturated lipids, which could be used to identify the location of C=C bonds in lipids with MS. However, this study and other similar work didn't solve the problem for identifying the configuration of C=C bonds in unsaturated lipids. In this work, we developed a bifunctional photocatalytic reaction system that could enable the identification of both the location and cis-trans configurations of C=C bonds in unsaturated lipids. This reaction has two different pathways: [2+2] photocycloaddition of carbonyl group with C=C, and cis-trans photoisomerization of C=C. The photocycloaddition reaction could be used to identify the location of C=C with tandem mass spectrometry (MS), whereas the photoisomerization reaction could tell the configurations of C=C bonds when combined with LC-MS. We have modified the description about comparing the difference of our previous study with the current one in introduction section. Please see lines 51-53 on page 3 in the manuscript.

4. Other comments:

Figure 1: The overall LCMS workflow for identifying *cis/trans* isomers is not clear.

Reply: The Figure 1 has been modified, and the schematic diagram for identifying *cis/trans* isomers was added. The more detailed workflow for the simultaneous identification of positional and *cis/trans* isomers by using the bifunctional photocatalytic reaction system and LC-MS/MS was shown in Figure 5 on page 15 in the manuscript.

Figure 4b: m/z 187 should be labeled as $[MBF+Na]^+$

Reply: The Figure 4b has been modified as the suggestion of the reviewer.

Figure 4d: m/z 451, 504 and 625 do not appear to be correctly labeled based on the scale of the x-axis. For example, m/z 451 is far above the mid-point between 400 and 500. It cannot be 451. 625 is far too close to 600.

Reply: The mistakes in Figure 4d have been corrected.

The gradient scale bar shown in Figure 5g does not show the whole range of colors. Nor is it clear what is being shown without a title for the gradient scale bar. It is a *trans/cis* ratio or something else? This should be added to the heat gradient scale bar.

Reply: Thanks for the suggestion. Figure 5g shows the EIC peak area ratios of lipids *trans*-isomers to their *cis*-counterpart from the four bacterial samples. The greatest ratio value in the scale bar was set at 2 to make color change more distinguishable, because most of the values are less than 2. In order to make it clearer, we added the values into the boxes which have ratio values more than 2. In addition, we plotted another heat map that shows the whole range of colors and added it into the Supplementary Information. The legend of the color scale bar was also added. Please see Figure 5g in the manuscript and Figure S24 in the Supplementary Information.

Supporting Information: Mass accuracies should be tabulated for all identified fragments and added to the SI.

Reply: The masses of all the fragment ions of the identified lipids in bacterial samples have been added into the Supplementary Datasheet.

REVIEWERS' COMMENTS

Reviewer #1 (Remarks to the Author):

The authors have successfully incorporated the majority of the requested additions and corrections, therefore, publication is recommended.

Reviewer #2 (Remarks to the Author):

The authors have corrected and improved the manuscript according to my initial review. The additional data, notably the additional of complementary GC analysis, strengthen the manuscript by confirming some of the more unusual fatty acid assignments that have been made.

The new method does a good job for fatty acid analysis where both double bond position and stereochemistry can be confidently assigned. With assignments now confirmed by conventional methods. Nevertheless the conventional methods can also do this. For complex lipids examples are shown where both double bond position and stereochemistry are assigned (e.g., Figure 5(C) and (G)). These suggest the method does extend beyond simple FAs to phospholipids and glycerolipids. Based on this contribution it should be published in its present form.

Reviewer #3 (Remarks to the Author):

The authors have addressed my original comments in a satisfactory manner, and I support publication of the manuscript in Nature Communications.